# Pharmacological rescue of impaired mitophagy in Parkinson's disease-related LRRK2 G2019S knock-in mice

**Francois Singh[1], Alan R Prescott[2], Philippa Rosewell[1], Graeme Ball[2], Alastair D Reith[3], Ian G Ganley[1]***

[1]MRC Protein Phosphorylation and Ubiquitylation Unit, University of Dundee, Dundee, United Kingdom; [2]Dundee Imaging Facility, School of Life Sciences, University of Dundee, Dundee, United Kingdom; [3]Novel Human Genetics Research Unit, GlaxoSmithKline Pharmaceuticals R&D, Stevenage, United Kingdom

**Abstract** Parkinson's disease (PD) is a major and progressive neurodegenerative disorder, yet the biological mechanisms involved in its aetiology are poorly understood. Evidence links this disorder with mitochondrial dysfunction and/or impaired lysosomal degradation – key features of the autophagy of mitochondria, known as mitophagy. Here, we investigated the role of LRRK2, a protein kinase frequently mutated in PD, in this process in vivo. Using mitophagy and autophagy reporter mice, bearing either knockout of LRRK2 or expressing the pathogenic kinase-activating G2019S LRRK2 mutation, we found that basal mitophagy was specifically altered in clinically relevant cells and tissues. Our data show that basal mitophagy inversely correlates with LRRK2 kinase activity in vivo. In support of this, use of distinct LRRK2 kinase inhibitors in cells increased basal mitophagy, and a CNS penetrant LRRK2 kinase inhibitor, GSK3357679A, rescued the mitophagy defects observed in LRRK2 G2019S mice. This study provides the first in vivo evidence that pathogenic LRRK2 directly impairs basal mitophagy, a process with strong links to idiopathic Parkinson's disease, and demonstrates that pharmacological inhibition of LRRK2 is a rational mitophagy-rescue approach and potential PD therapy.

**\*For correspondence:**
i.ganley@dundee.ac.uk

**Competing interests:** The authors declare that no competing interests exist.

## Introduction

Parkinson's disease (PD) is the second most common neurodegenerative disorder, affecting 1–2% of the population over 60 years old, and 4% above 85 years of age (*Coppedè, 2012*). The main symptoms of PD are muscle rigidity, bradykinesia, resting tremor, and postural instability and may be accompanied by sleep disorders, anosmia, depression, and dementia (*Sironi et al., 2020*). It is characterised by a progressive and selective degeneration of dopaminergic (DA) neurons of the substantia nigra pars compacta (SNpc). Currently, there are no treatments available that modify the course of neurodegenerative decline. Although this disease is mostly sporadic, about 15% of cases appear to be inherited and in support of this, 20 genes implicated in PD have been identified from familial genetic studies while ~90 loci have been identified from PD-GWAS (*Deng et al., 2018*). The exact causes of PD are currently unknown but some evidence strongly links impaired mitochondrial and lysosomal function to disease pathology (*Sironi et al., 2020*).

Mutations in *PARK8*, encoding for LRRK2 (Leucine-Rich Repeat Kinase 2), are the most frequently reported cause of PD (*Bouhouche et al., 2017*). The most common mutation associated with PD is the substitution of glycine at position 2019 of LRRK2 to serine (G2019S), representing 4% of familial and 1% of sporadic cases (*Lill, 2016*). LRRK2 is a large multidomain protein with two catalytic domains: a Ras of complex (ROC) GTPase domain that is able to bind GTP and hydrolyse it, and a kinase domain that utilises a subset of Rab GTPases as substrates (*Steger et al., 2016*). Importantly,

all the segregating mutations associated with PD are located in the catalytic core. A mutation in the ROC/COR domain, such as the R1441C/G/H or the Y1699C mutation, leads to decreased GTPase activity and elevated kinase activity (*Rudenko and Cookson, 2014*). Mutations in the kinase domain, such as the G2019S or the I2020T, also lead to an elevated kinase activity. Hence, enhanced kinase activity appears to be a common factor in pathogenic LRRK2 mutations. Although the function of LRRK2 within cells is currently unknown, mounting evidence implicates a role in membrane trafficking (*Steger et al., 2016*; *Pfeffer, 2018*; *Hur et al., 2019*).

Macroautophagy is a membrane trafficking pathway that delivers intracellular components to the lysosome for degradation (*Yu et al., 2018*). These components can include whole organelles such as mitochondria. The autophagic turnover of mitochondria is termed mitophagy, which acts as a mitochondrial quality control mechanism that allows the selective degradation of damaged or unnecessary mitochondria (*Rodger et al., 2018*; *Montava-Garriga and Ganley, 2020*). Mitophagy itself has strong links to PD following the landmark discoveries that PINK1 and Parkin, two other genes mutated in familial PD, sequentially operate to initiate mitophagy in response to mitochondrial depolarisation in cell lines (*Narendra et al., 2008*; *Koyano et al., 2014*; *Pickrell and Youle, 2015*; *Yamano et al., 2018*). However, when this pathway becomes relevant in vivo, and under what physiological conditions, is unclear especially given that PINK1 and Parkin are not required for regulation of mitophagy under normal, or basal, conditions (*McWilliams et al., 2018a*; *McWilliams et al., 2018b*; *Lee et al., 2018*). Indeed, our understanding of the detailed mechanisms regulating basal mitophagy remains elusive.

In this study, we sought to define the physiological link between mitochondrial turnover and LRRK2 in relation to PD. We utilised our previously published mouse reporter models to study mitophagy (*mito*-QC) and autophagy (*auto*-QC; Fig. 1A, D and *McWilliams et al., 2018a*; *McWilliams et al., 2016*; *McWilliams et al., 2019*) in either LRRK2 knockout mice, or knock-in mice harbouring the pathogenic LRRK2 G2019S mutation. Whilst we found minimal impact of LRRK2 on general autophagy (macroautophagy), we observed that the LRRK2 G2019S activation-mutation was associated with reduced mitophagy in specific tissues, including dopaminergic neurons and microglia within the brain. In contrast, knockout of LRRK2 resulted in increased mitophagy. Taken together, these data imply that LRRK2 kinase activity inversely correlates with basal mitophagy levels. In support of this, we found that that treatment of cells or animals with the potent and selective CNS penetrant LRRK2 kinase inhibitor, GSK3357679A (*Tasegian et al., 2021*; Ding, 2021, in preparation), rescued these LRRK2 G2019S-associated mitophagy defects and enhanced mitophagy in dopamine neurons and microglia in the brains of genotypically normal mice. Our results identify a physiological role for LRRK2 in the regulation of basal mitophagy in vivo and underline the potential value of pharmacological inhibition of LRRK2 as a potential therapeutic strategy to ameliorate aspects of Parkinson's disease driven by mitochondrial dysfunction.

## Results

### The pathogenic G2019S LRRK2 mutation impairs basal mitophagy but not autophagy in vitro

To investigate the physiological role of LRRK2 in regulating autophagy we utilised two previously validated and highly similar mouse reporter models (*McWilliams et al., 2018a*; *McWilliams et al., 2016*; *McWilliams et al., 2019*). These transgenic reporter models rely on constitutive expression of a tandem mCherry-GFP tag from the *Rosa26* locus. In the *mito*-QC model, which monitors mitophagy, the tandem tag is localised to mitochondria (by an outer mitochondrial targeting sequence derived from residues 101–152 of the protein FIS1). In the *auto*-QC model, which monitors general (macro)autophagy, the tandem tag is localised to autophagosomes (by conjugation to the N-terminus of MAP1LC3b). For both models, when a mitochondrion or autophagosome is delivered to lysosomes, the low lysosomal luminal pH is sufficient to quench the GFP signal, but not that from mCherry. Hence, the degree of mitophagy or general autophagy can be determined by the appearance of mCherry-only puncta, which represent mito/autolysosomes (*Figure 1A and D*). Given that mitophagy is a form of autophagy, the use of both models allows us to monitor the specificity of autophagy in vivo. A large disruption of autophagy in general will also influence mitophagy, whereas

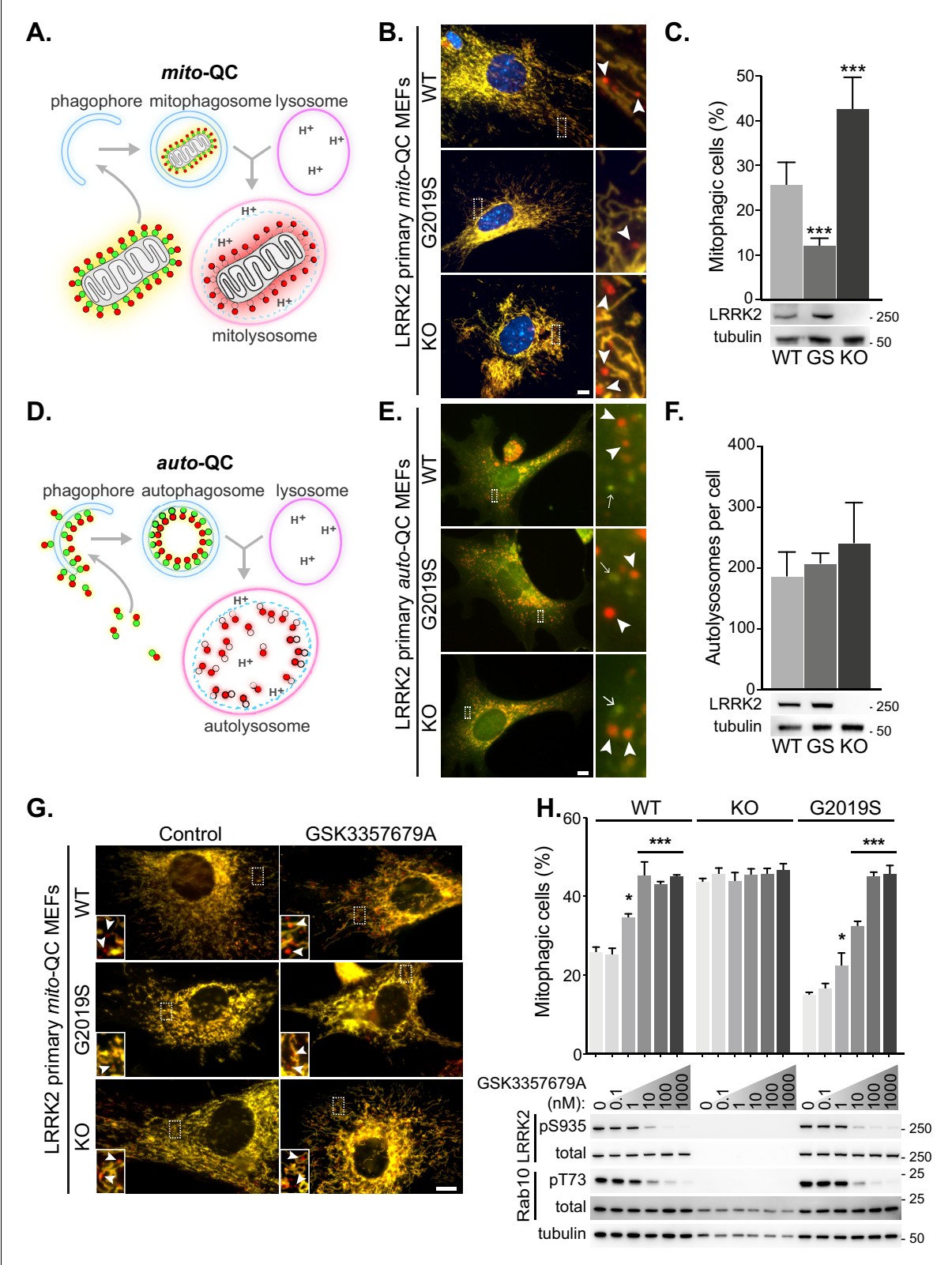

**Figure 1.** LRRK2 kinase activity impairs basal mitophagy in vitro. (A) Schematics of the *mito*-QC reporter in mouse model. (B) Representative images of *mito*-QC primary MEF cultures established from LRRK2 WT, LRRK2 G2019S, and LRRK2 KO embryos. Boxed area is magnified on the right and arrowheads indicate examples of mitophagy (mCherry-only mitolysosomes). (C) Mitophagy quantitation of data shown in B from six to nine independent experiments, as described in Materials and methods. Below is representative immunoblot showing LRRK2 protein expression. (D)
*Figure 1 continued on next page*

*Figure 1 continued*

Schematics of the *auto*-QC reporter in mouse model. (**E**) Representative images of *auto*-QC primary MEF cultures established from LRRK2 WT, LRRK2 G2019S, and LRRK2 KO embryos. Boxed area is magnified on the right and arrowheads indicate examples of autolysosomes and arrows highlight autophagosomes. (**F**) Quantitation of data shown in E from four to six independent experiments. (**G**) Representative images of *mito*-QC primary MEFs treated with control (DMSO) or 10 nM GSK3357679A. Boxed area is magnified on bottom left and arrowheads indicate examples of mitolysosomes. (**H**) Quantitation of mitophagy shown in G from three to seven independent experiments. Corresponding immunoblot of indicated proteins is shown below. Scale bars, 10 µm. Overall data is represented as mean +/- SEM. Statistical significance is displayed as *p<0.05, and ***p<0.001.

The online version of this article includes the following source data and figure supplement(s) for figure 1:

**Source data 1.** Numerical data for *Figure 1*.
**Source data 2.** Western blot raw data files for *Figure 1C*.
**Source data 3.** Western blot raw data files for *Figure 1F*.
**Source data 4.** Western blot raw data files for *Figure 1H*.
**Figure supplement 1.** *Lrrk2* genotype does not impact stimulated mitophagy and autophagy, yet multiple LRRK2 kinase inhibitors enhance mitophagy under basal conditions.
**Figure supplement 1—source data 1.** Numerical data for *Figure 1—figure supplement 1*.
**Figure supplement 1—source data 2.** Western blot raw data files for *Figure 1—figure supplement 1E*.

a block in mitophagy, which likely represents a small fraction of the total autophagy occurring at any one time, will tend to have little influence on the total autophagic levels.

To investigate the effect of LRRK2 kinase activity on mitophagy, we first isolated and cultured primary mouse embryonic fibroblasts (MEFs) derived from wild-type mice (WT), mice homozygous for the Parkinson's disease-associated LRRK2 G2019S mutant, or mice homozygous for a LRRK2 knock-out (KO) variant; all of which were on a homozygous *mito*-QC reporter background (*Figure 1B and C*). A small degree of basal mitophagy was evident in all cell lines. However, we observed that LRRK2 G2019S KI mutant cells displayed significantly lower basal mitophagy levels, whereas the absence of LRRK2 (KO) led to an increase of this process. Interestingly, our data suggests that LRRK2 predominantly influences basal mitophagy as deferiprone (DFP) and long-term amino acid starvation in Earls Balanced Salt Solution (EBSS), both strong mitophagy inducers (*Allen et al., 2013*), increased mitophagy to a similar level across all genotypes (*Figure 1—figure supplement 1A and B*).

We next investigated general autophagy using the LRRK2 mouse lines mentioned above on the homozygous *auto*-QC background. In contrast to mitophagy, in isolated primary MEFs we noticed no significant difference in the number of mCherry-only autolysosomes across all the *Lrrk2* genotypes under basal conditions (*Figure 1E and F*). We also analysed amino acid starvation-induced autophagy, by EBSS incubation. A robust autophagy response was observed in all cells and as with basal autophagy, the *Lrrk2* genotype failed to significantly alter this large increase in autolysosomes (*Figure 1—figure supplement 1C*).

## LRRK2 kinase inhibitors correct the G2019S mitophagy defects in vitro

Using genetics, our observations show that LRRK2 kinase activity inversely correlates with mitophagy in vitro. If this is the case, then pharmacological inhibition of LRRK2 kinase activity should also increase mitophagy. Therefore, we aimed to investigate if the mitophagy deficit observed in the G2019S cells could be rescued with LRRK2-selective kinase inhibitors. To that end, we turned to GSK3357679A, a novel pyrrolopyrimidine LRRK2 kinase inhibitor that exhibits excellent cellular potency, selectivity, oral bioavailability and pharmacokinetics/pharmacodynamics correlation in animal studies (*Tasegian et al., 2021*; Ding, 2021, in preparation). We tested GSK3357679A in primary *mito*-QC MEFs and observed a dose-dependent effect on mitophagy with a maximal stimulation achieved at a concentration of 10 nM in WT cells (*Figure 1G and H*). The level of stimulation was equivalent to the level of mitophagy observed in LRRK2 KO cells. Importantly, GSK3357679A failed to alter mitophagy in the absence of LRRK2, demonstrating that its mitophagy-enhancing properties are dependent on LRRK2. In the G2019S cells, we observed a reduced response, with a maximal effect on mitophagy reached at 100 nM and this may reflect the increased kinase activity of this mutant (*Nichols et al., 2009*). Western blotting analysis confirmed that GSK3357679A potently inhibited LRRK2 kinase activity in a dose-dependent manner, as indicated by decreased

phosphorylation of its substrate Rab10 at threonine 73 (*Steger et al., 2016*), as well as reduced LRRK2 S935 phosphorylation (an indirect measure of LRRK2 activity (*Ito et al., 2016*), *Figure 1H*).

To further support a role for LRRK2 kinase activity in negatively regulating basal mitophagy, we utilised two additional and structurally distinct tool LRRK2 kinase inhibitors, GSK2578215A (*Reith et al., 2012*) and MLi-2 (*Fell et al., 2015*), in primary *mito*-QC MEFs. As with GSK3357679A, both these compounds were able to inhibit LRRK2 in cells and increase mitophagy (*Figure 1—figure supplement 1D and E*). We do note that at high concentrations, MLi2 failed to stimulate mitophagy and this may be due to off-target effects, as at 20 nM it also inhibited mitophagy in the LRRK2 KO cells (*Figure 1—figure supplement 1D*). Thus, genetically and chemically, the data show that LRRK2 inhibition enhances basal mitophagy in cells and in these assays, GSK3357679A displayed a superior performance compared to other available LRRK2 kinase inhibitors.

## LRRK2 inhibition activates conventional mitophagy independently of the PINK1 pathway

Although *mito*-QC monitors the delivery of mitochondria to lysosomes, it does not give any mechanistic insight as to how this occurs. Given that multiple mitophagy-like pathways exist, which can be dependent or independent of the canonical macroautophagy machinery (*Montava-Garriga and Ganley, 2020*), we decided to investigate this further. LC3 is the classical autophagosome marker and if mitophagy is occurring via canonical autophagy then LRRK2 inhibition should result in increased co-localisation between mitochondria and LC3. To test this, we took advantage of our primary *auto*-QC MEFs and co-stained these for the mitochondrially localised ATP synthase subunit beta (ATPB, *Figure 2A*). Treatment of cells with GSK3357679A significantly increased co-localisation of mCherry-GFP-LC3 with ATPB, both at the autophagosome stage (GFP and mCherry-positive, *Figure 2B*) and autolysosome stage (mCherry-only positive, *Figure 2C*). This implies LRRK2 inhibition uses a canonical autophagy pathway to drive mitophagy. To confirm the requirement for the LC3 conjugation machinery in this form of mitophagy, we expressed the *mito*-QC reporter in the previously reported macroautophagy-deficient ATG5 KO MEFs (*Kuma et al., 2004*). In the matched WT MEFs, GSK3357679A was able to induce mitophagy in a similar manner to the primary *mito*-QC MEFs; however, mitophagy induction was blocked in the ATG5 KO MEFs (*Figure 2D*). As a final, reporter-independent method to confirm mitophagy, we carried out transmission electron microscopy (TEM, *Figure 2E and F*). To aid in this process, we first immortalised the primary *mito*-QC MEFs, to remove their time limitation, and confirmed that LRRK2 inhibition could still induce mitophagy (*Figure 2—figure supplement 1A–C*). Partially degraded mitochondrial structures could clearly be identified within autolysosomes (representative images shown in *Figure 2E*) and when quantified, a significant 1.3-fold increase was detected in the GSK3357679A-treated samples. Taken together, the data show that LRRK2 inhibition stimulates mitophagy through canonical ATG5-dependent autophagy and confirms that the *mito*-QC reporter is measuring a conventional mitophagy pathway in this instance.

The most widely studied mitophagy pathway involves the activation of PINK1 and Parkin that occurs following mitochondrial depolarisation (*Montava-Garriga and Ganley, 2020*). Given that mutations in PINK1, Parkin and LRRK2 can all lead to PD, it was important to determine if LRRK2 inhibition resulted in PINK1-dependent mitophagy. To test this, we used previously generated primary MEFs derived from *mito*-QC x PINK1 KO mice (*McWilliams et al., 2018a*). Treatment of littermatched WT and PINK1 KO *mito*-QC MEFs with GSK3357679A resulted in a comparable increase in mitophagy regardless of the presence or absence of PINK1 (*Figure 2G and H*). To support the PINK1-independent nature of this pathway we immunoblotted for PINK1-dependent phospho-ubiquitin. We observed no detectible increase in phospho-ubiquitin levels following LRRK2 inhibition, which contrasted with that observed following mitochondrial depolarisation with CCCP treatment (*Figure 2I*). To further examine the relationship between PINK1/Parkin-dependent mitophagy and LRRK2 inhibition, we sought to directly monitor Parkin-dependent mitophagy. As high levels of Parkin are needed to observe PINK1/Parkin-dependent mitophagy, we next overexpressed HA-tagged Parkin and induced mitophagy with CCCP treatment (*Figure 2—figure supplement 1D–F*). *mito*-QC clearly detected a large and significant increase in mitophagy following mitochondrial depolarisation, but this was unaffected by GSK3357679A, as were phospho-ubiquitin levels.

The lack of involvement of PINK1 and Parkin in the observed mitophagy could imply that mitochondrial depolarisation is not a major trigger for this pathway. We thus explored mitochondrial

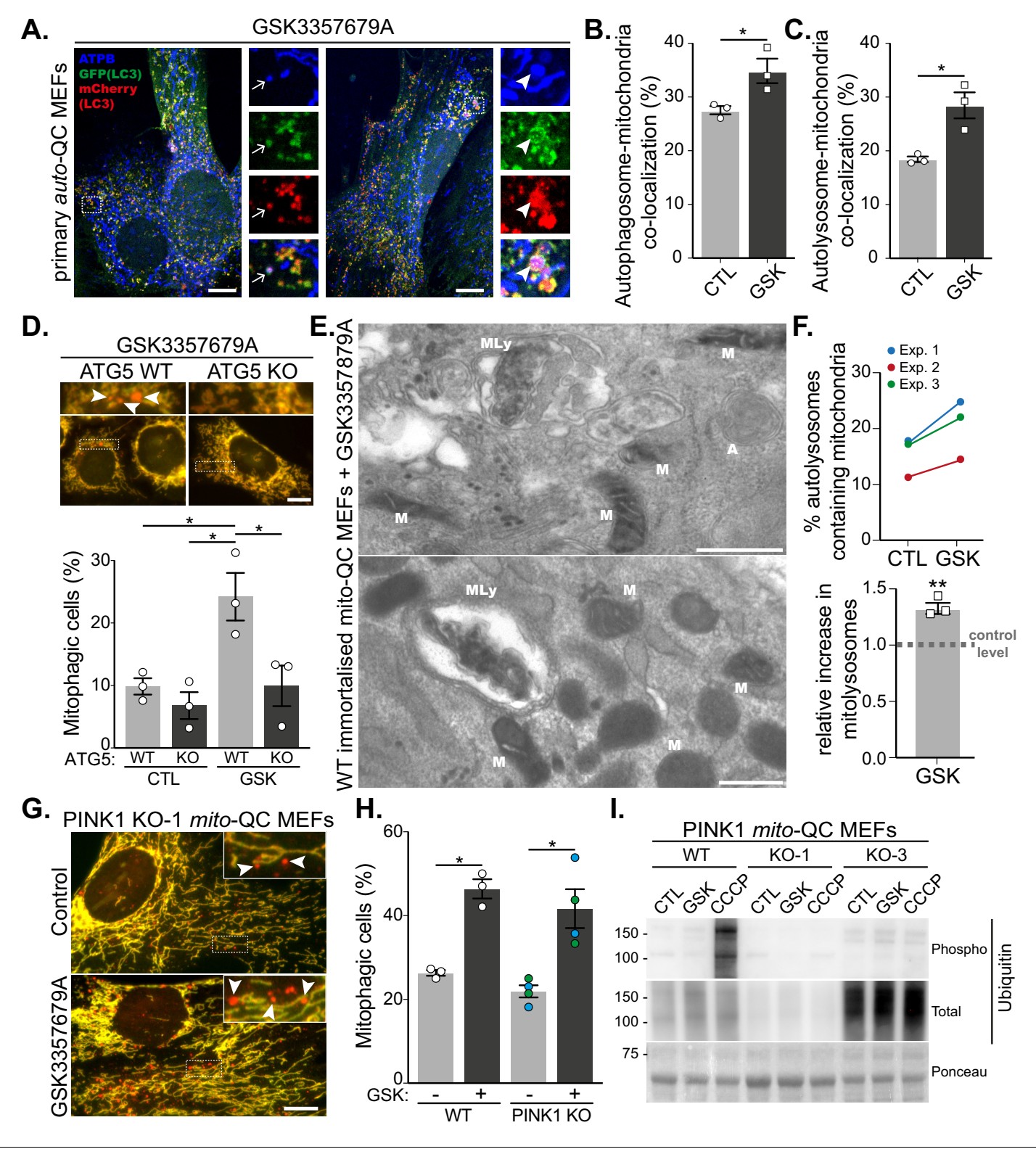

**Figure 2.** LRRK2-dependent mitophagy relies on the general autophagy machinery but is independent of the PINK1 pathway. (**A**) Representative images of *auto*-QC primary MEF cultures treated with 10 nM GSK3357679A for 24 hr and immunolabelled with ATPB. Arrows show examples of autophagosomes containing mitochondria, and arrowheads show examples of autolysosomes containing mitochondria. Scale bars, 10 μm. (**B**) Quantitation of autophagosomes containing mitochondria shown in (**B**) or autolysosomes containing mitochondria shown in (**C**) from three independent

*Figure 2 continued on next page*

*Figure 2 continued*

experiments. (D) Representative images of WT and ATG5 KO *mito*-QC immortalised MEF cultures treated with 10 nM GSK3357679A for 24 hr. Corresponding quantitation from three independent experiments is shown below. Scale bars, 10 μm. (E) Representative transmission electron microscopy images from immortalised MEFs treated with 10 nM GSK3357679A for 24 hr. MLy: Mitolysosome; M: Mitochondria; A: autophagosome. Scale bars, 500 nm. (F) Quantitation of data in E from three independent experiments. (G) Representative images of PINK1 KO *mito*-QC primary MEF cultures treated with/without with 10 nM GSK3357679A for 24 hr. Scale bars, 10 μm. (H) Quantitation of the of mitophagic cells from data shown in G from 3 (4 for KO) independent experiments. (I) Immunoblots of the indicated proteins from WT and PINK1 KO primary MEFs (KO-1 and KO-3 are derived from different embryos) treated with/without 10 nM GSK3357679A or 10 μM CCCP for 24 hr. Overall data is represented as mean +/- SEM. Statistical significance is displayed as *p<0.05 and **p<0.01.

The online version of this article includes the following source data and figure supplement(s) for figure 2:

**Source data 1.** Numerical data for *Figure 2*.
**Source data 2.** Western blot raw data files for *Figure 2I*.
**Figure supplement 1.** LRRK2-regulated mitophagy does not disrupt Parkin-dependent mitophagy and does not globally affect mitochondrial function.
**Figure supplement 1—source data 1.** Numerical data for *Figure 2—figure supplement 1*.
**Figure supplement 1—source data 2.** Western blot raw data files for *Figure 2—figure supplement 1A*.
**Figure supplement 1—source data 3.** Western blot raw data files for *Figure 2—figure supplement 1F*.

function in general in response to LRRK2 inhibition and GSK3357679A treatment. Firstly, we noticed no obvious changes to mitochondrial morphology and ultrastructure, as observed using TEM (*Figure 2—figure supplement 1G*). Secondly, using high-resolution respirometry, we measured mitochondrial oxygen consumption. We first investigated if GSK3357679A treatment directly affected the mitochondrial respiratory chain. Thus, we activated the NADH- and succinate-linked pathway and injected sequentially incremental doses of GSK3357679A. This had no direct effect on the mitochondrial respiratory chain (*Figure 2—figure supplement 1H*). Secondly, we evaluated the chronic effects of LRRK2 inhibition by incubating immortalised MEFs for 24 hr with 10 nM GSK3357679A (*Figure 2—figure supplement 1I*). Despite an increase in mitophagy (*Figure 2—figure supplement 1C*), we did not observe any difference in mitochondrial respiration. Furthermore, we did not observe any change in substrate preference. Thus, LRRK2 inhibition does not globally impact mitochondrial form or function at this timepoint. However, given that the fraction of mitochondria targeted for mitophagy is likely small compared to the total pool, we cannot rule out that this population is functionally impaired.

## Mutation of LRRK2 in vivo alters mitophagy in specific cell populations within the brain

Given the effects of LRRK2 kinase activity on mitophagy in vitro, we next sought to use our mouse lines to investigate this in vivo. PD is primarily a neurodegenerative disorder, so we first explored mitophagy in the brain. We focussed on four cell populations: two neuronal populations linked to movement – dopaminergic (DA) neurons of the substantia nigra pars compacta (SNpc) and Purkinje neurons of the cerebellum; as well as in two glial cell populations – cortical microglia and cortical astrocytes. In midbrain, we identified SNpc DA neurons using tyrosine hydroxylase (TH) staining and found no difference in the number of DA neurons per field across the *Lrrk2* genotypes (*Figure 3A and B*). These cells are the mouse equivalent of the human population of DA neurons that degenerate in PD and we have previously found that they undergo substantial mitophagy (*McWilliams et al., 2018a*). Basal mitophagy was significantly enhanced in the LRRK2 KO neurons compared to WT, and although not statistically significant, mitophagy appeared reduced in DA neurons of LRRK2 G2019S KI mice compared to WT (*Figure 3A and C*). While we cannot say that mitophagy is significantly impaired in the G2019S DA neurons from this set of experiments using nine individual mice, we did however find a significant mitophagy reduction in G2019S DA neurons in a later set of experiments, with 10 mice per condition (Figure 6A and B). Taken together, these observations are similar to our earlier results in MEFs and showed that the presence of LRRK2 can impact mitophagy in this clinically relevant population of neurons within the midbrain. To determine if this effect is typical of neurons in general, we investigated mitophagy in another neuronal population involved in motor control, the Purkinje neurons. These cells were identified in cerebellar sections using immunostaining against the calcium sensor Calbindin-D28k. These cells are rich in mitochondria and as shown previously (*McWilliams et al., 2016*), they also undergo significant mitophagy (*Figure 3D*). Contrary to what

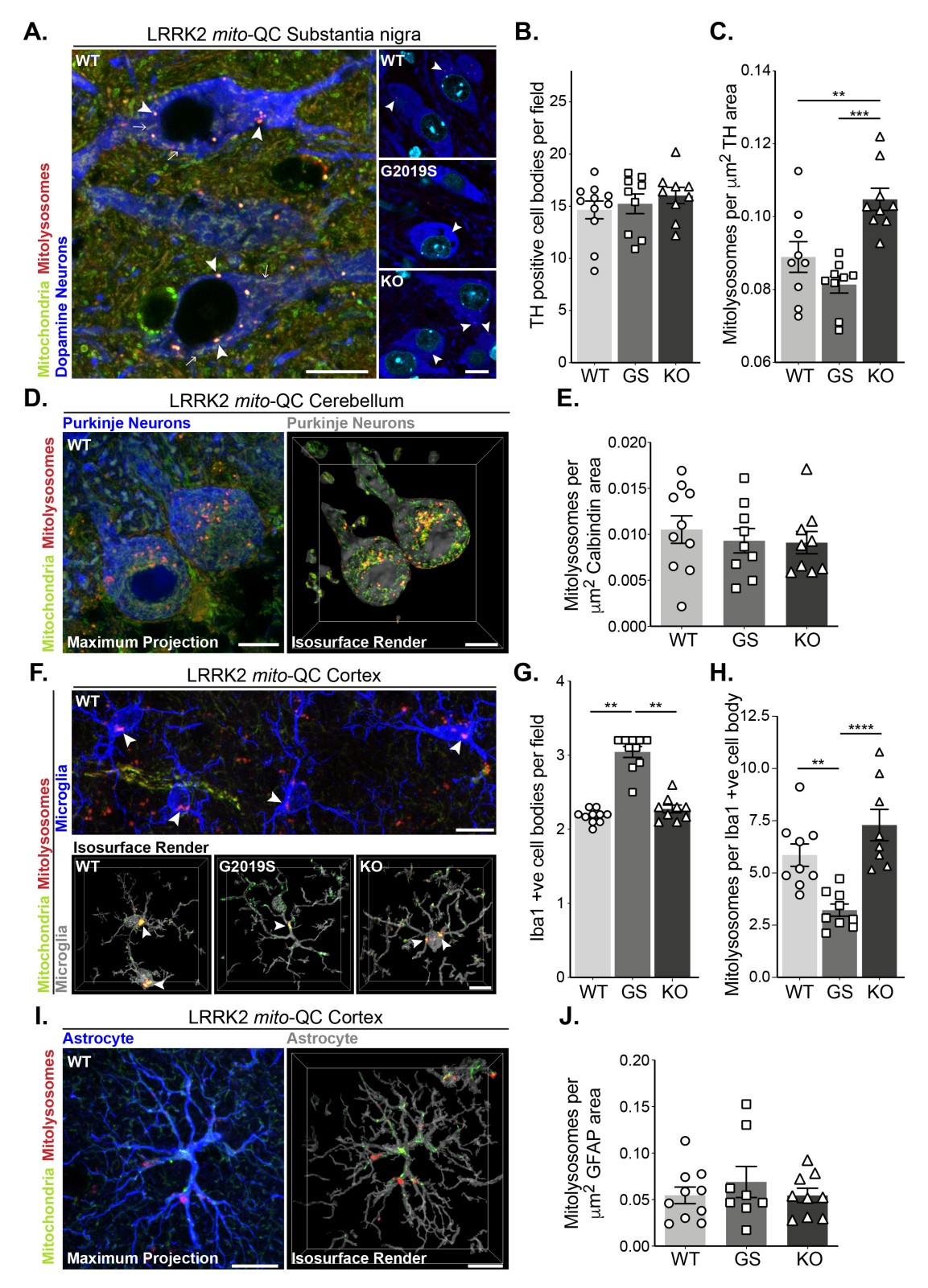

**Figure 3.** Mutation of LRRK2 in vivo alters brain mitophagy. (**A**) Representative image of tyrosine hydroxylase (TH) immunolabelled dopaminergic neurons within the substantia nigra pars compacta (SNpc) undergoing basal mitophagy in LRRK2 WT, LRRK2 G2019S, and LRRK2 KO *mito*-QC mice. Arrowheads show examples of mitolysosomes and arrows indicate mitochondria. (**B**) Quantitation of the number of TH positive cells per field of view using a 63x objective. (**C**) Quantitation of basal mitophagy per μm (*Sironi et al., 2020*) of TH staining, data points represent means from individual

*Figure 3 continued on next page*

*Figure 3 continued*
mice. (D) Representative maximal intensity projection and isosurface render of calbindin immunolabelled Purkinje neurons in *mito*-QC cerebellum sections, undergoing basal mitophagy. (E) Quantitation of basal mitophagy per µm (*Sironi et al., 2020*) of calbindin staining, data points represent means from individual mice. (F) Representative image and isosurface renders of Iba1 immunolabelled microglia undergoing basal mitophagy in cortical sections from LRRK2 WT, LRRK2 G2019S, and LRRK2 KO *mito*-QC mice. Arrowheads highlight mitolysosomes. (G) Quantitation of the number of Iba1-positive cells in the brain cortex per field of view using a 63x objective. (H) Quantitation of basal mitophagy per Iba1-positive cell body, data points represent means from individual mice. (I) Representative maximal intensity projection and isosurface render of GFAP immunolabelled astrocytes undergoing basal mitophagy in *mito*-QC cortical sections. (J) Quantitation of basal mitophagy per µm (*Sironi et al., 2020*) of GFAP staining, data points represent means from individual mice. Scale bars, 10 µm. Overall data is represented as mean +/- SEM. Statistical significance is displayed as *p<0.05, **p<0.01, and ***p<0.001.

The online version of this article includes the following source data and figure supplement(s) for figure 3:

**Source data 1.** Numerical data for *Figure 3*.
**Figure supplement 1.** Macroautophagy in the brain is unaltered by *LRRK2* genotype.
**Figure supplement 1—source data 1.** Numerical data for *Figure 3—figure supplement 1*.

we observed in SNpc DA neurons, no statistical difference in mitophagy in Purkinje cells was found between any group (*Figure 3E*).

As we observed neuron-specific alterations of mitophagy, we next examined the effect of *Lrrk2* genotype in two distinct populations glial cells within the cortex. Immune-related microglia were identified by Iba1 (ionised calcium-binding adapter molecule 1, *Figure 3F*). Notably, we observed an enhanced presence of microglial cells in the cortex of G2019S animals when compared to WT or KO mice (*Figure 3G*). We do not yet understand the nature of this increase and further work will be needed to determine if there are simply more microglia in the G2019S mice, or an increased movement of cells to this area of the brain. Regardless, when mitophagy quantitation was normalised for cell number (mitolysosomes per Iba1-positive cell body per field), we found a significant decrease in basal mitophagy in G2019S microglia compared to WT, as well as an increase in mitophagy levels in KO cells (*Figure 3H*). Thus, as with DA neurons, LRRK2 can impacts basal mitophagy in microglia. In contrast, cortical astrocytes, stained with glial fibrillary acidic protein (GFAP, *Figure 3I*), did not show any observable difference in mitophagy across *Lrrk2* genotypes (*Figure 3J*).

In contrast to the *Lrrk2* genotype effects on mitophagy in DA neurons and microglia, analysis of *auto*-QC mouse brains indicated no change in general macroautophagy in these cell types (*Figure 3—figure supplement 1*). Thus, the LRRK2 G2019S mutation is not causing a major disruption in neuronal autophagy but does influence basal mitophagy levels.

## Mutation of LRRK2 in vivo also alters mitophagy in peripheral organs with high LRRK2 expression

We next assessed mitophagy levels in the lungs, a tissue in which the levels of LRRK2 are known to be elevated (*Uhlén et al., 2005*; *Uhlen et al., 2010*). Basal mitophagy across the whole lung was evident in all genotypes and in a similar fashion to MEFs, DA neurons and microglia, mitophagy was reduced in G2019S mice and enhanced (over 2-fold relative to WT) in KO mice (*Figure 4A and B*), Consistent with this, mitochondrial content was decreased when comparing KO to G2019S (although no significant increase of this parameter was detected compared to WT, see *Figure 4—figure supplement 1A*). As previously reported (*Baptista et al., 2018*; *Plowey et al., 2008*), we observed enlarged type II pneumocytes with considerable vesicular-like structures in all the animals of the LRRK2 KO group that is attributable to the accumulation of large lamellar bodies, which are secretory lysosomes responsible for surfactant release. We confirmed the nature of these structures as enlarged lamellar bodies by filipin staining lung sections for cholesterol, a component found in surfactant (*Figure 4—figure supplement 1B*).

The tissue reported to have the highest LRRK2 expression is the kidney (*Uhlén et al., 2005*; *Uhlen et al., 2010*). We first investigated mitophagy in the kidney cortex, where we had previously shown the proximal tubules to be a major site of mammalian mitophagy (*McWilliams et al., 2016*). LRRK2-dependent mitophagy changes in the kidney were much lower in magnitude compared to the lung, yet there was a small decrease in G2019S-expressing tissue (*Figure 4C and D*). However, we do note that mitophagy is 10-fold higher in this region compared to lung, which may mask relatively small changes conferred by *Lrrk2* genotypes.

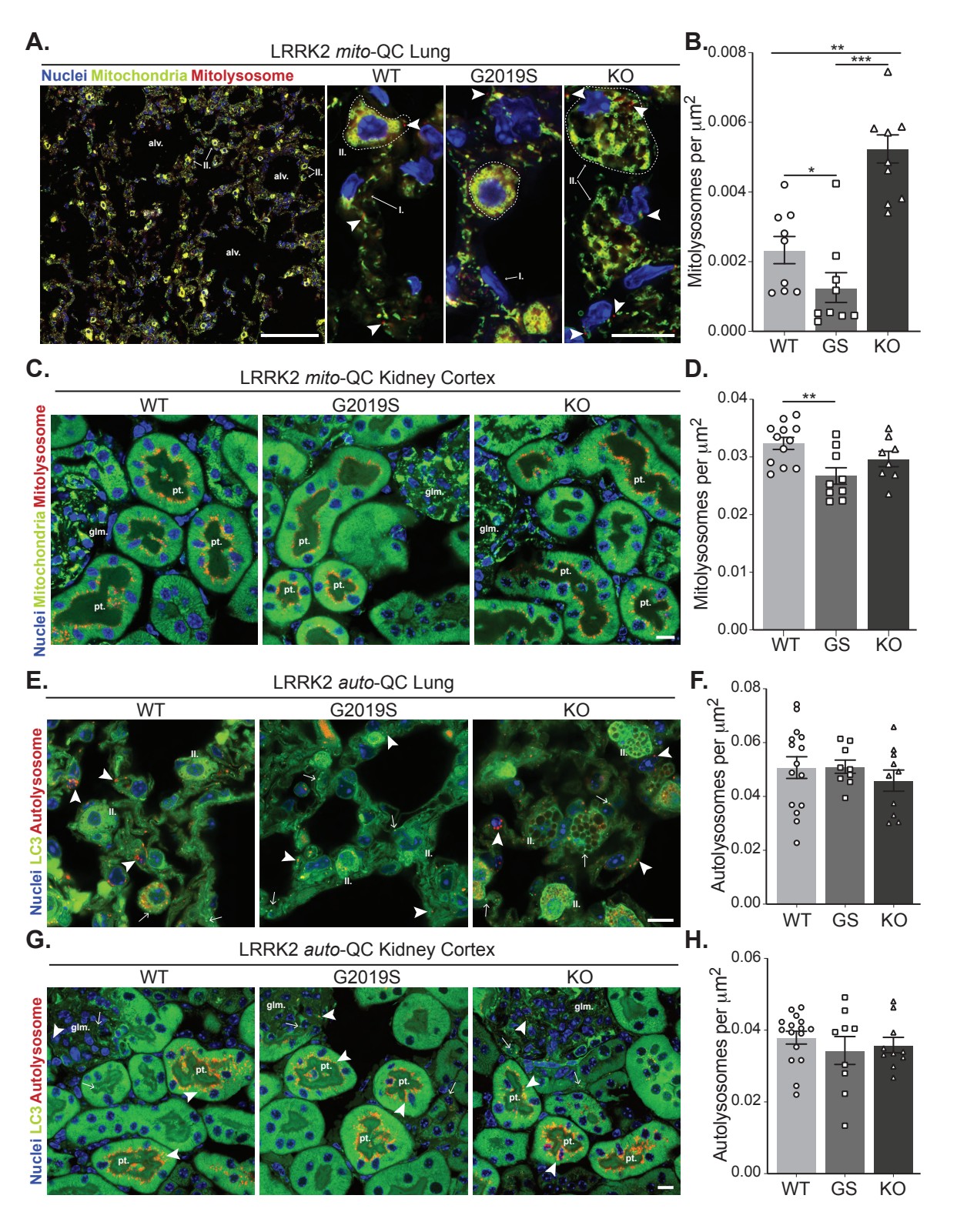

**Figure 4.** Effects of *Lrrk2* genotype on basal mitophagy and macroautophagy in selected peripheral tissues. (**A**) Representative tile scan and images of *mito*-QC lungs from LRRK2 WT, LRRK2 G2019S, and LRRK2 KO mice. Arrows on the tile scan highlight Type II pneumocytes (II.), alv. show alveoli. Arrowheads in the higher magnification images indicate examples of mitolysosomes and circled cells correspond to Type II pneumocytes (II.) and I. indicates Type I pneumocytes. (**B**) Quantitation of lung basal mitophagy from data shown in D, data points represent means from individual mice. (**C**)

*Figure 4 continued on next page*

*Figure 4 continued*

Representative images of *mito*-QC kidney cortex from LRRK2 WT, LRRK2 G2019S, and LRRK2 KO mice. glm. indicates glomeruli and pt. indicates proximal tubule examples. (D) Quantitation of kidney basal mitophagy from data shown in F, data points represent means from individual mice. (E) Representative images of *auto*-QC lungs from LRRK2 WT, LRRK2 G2019S, and LRRK2 KO mice. Arrowheads highlight autolysosomes, arrows indicate autophagosomes, and II. indicates Type II pneumocytes. (F) Quantitation of lung autophagy from data shown in C, data points represent means from individual mice. (G) Representative images of *auto*-QC kidney cortex from LRRK2 WT, LRRK2 G2019S, and LRRK2 KO mice. Arrowheads and, arrows as in C, glm. indicates glomeruli, and pt. indicates proximal tubules. (H) Quantitation of kidney autophagy from data shown in E, data points represent means from individual mice. Scale bars, Tile scan in A: 100 µm, Other pictures:10 µm. Overall data is represented as mean +/- SEM. Statistical significance is displayed as **p<0.01, and ***p<0.001.

The online version of this article includes the following source data and figure supplement(s) for figure 4:

**Source data 1.** Numerical data for *Figure 4*.
**Figure supplement 1.** Macroautophagy in the lungs and kidney cortex of LRRK2 mice.
**Figure supplement 1—source data 1.** Numerical data for *Figure 4—figure supplement 1*.

We next studied in vivo genotype effects on autophagy, using the same conditions and organs as for the *mito*-QC reporter. Consistent with brain and MEF data, no significant difference in the number of autolysosomes was observed in the lungs of *auto*-QC reporter mice (*Figure 4E and F*). Again, enlarged type II pneumocytes were observed in LRRK2 KO animals (*Figure 4E*). Likewise, in the kidney cortex, we did not detect an effect of *Lrrk2* genotype on autolysosomes (*Figure 4G and H*). We also note that no major difference was seen in the number of autophagosomes across both lung and kidney (*Figure 4—figure supplement 1C and D*). Taken together, these data suggest that the *Lrrk2* genotype does not majorly affect all autophagy pathways but predominantly impacts basal mitophagy, both in vitro and in vivo.

## GSK3357679A corrects the G2019S mitophagy defect in vivo

We next sought to determine if we could pharmacologically rescue the observed mitophagy defects in vivo. For this purpose, we utilised GSK3357679A – the pharmacodynamic characteristics of which have been shown to be suitable for extended oral dosing studies in rodents (*Tasegian et al., 2021*; Ding, 2021, in preparation). We administered *mito*-QC WT, G2019S, and LRRK2 KO mice with GSK3357679A via oral gavage every 12 hr for a total of four doses. During this period, we observed no effect of GSK3357679A on body weight in any genotype (*Figure 5—figure supplement 1A*). Tissues were then harvested 2 hr post the final dose. We focused our analyses on tissues where our previous analyses of *Lrrk2* genotypic variants suggested a LRRK2-dependent role in mitophagy, namely the brain and lung as well as the kidney.

Initially focussing on the brain, LRRK2 inhibition was confirmed in tissue lysates of GSK3357679A dosed mice by phosphosite immunoblotting of LRRK2 and its substrate Rab12 (*Figure 5A*). GSK3357679A decreased the phosphorylation of LRRK2 on S935 and the phosphorylation of Rab12 on S106, in both WT and G2019S mice (*Figure 5A and B*). Interestingly, the level of Rab12 phosphorylation was higher in G2019S brains, which could reflect the higher kinase activity that is associated with this mutation.

In the same tissue lysates, we also immunoblotted for mitochondrial markers (*Figure 5A and C*). We did not find any significant difference in the levels of the matrix-localised HSP60 or the outer membrane-localised TOMM20, in either WT or G2019S mice. We think the failure to see any increase in these markers in the G2019S brain likely reflects the low degree of basal mitophagy in general and that we are analysing whole brain tissue that will contain cells that are unresponsive to changes in LRRK2 activity (see *Figure 3D and I*). However, in the lysates analysed from two LRRK2 KO mice, both these proteins were reduced. This observation is consistent with increased mitophagy in LRRK2 KO cells, and given the data in G2019S brains, implies that the total loss of LRRK2 kinase activity has a relatively larger mitophagy effect than that the smaller activity change caused by the pathogenic mutation. Of potential relevance, we found that the mitochondrial biogenesis marker PGC-1α was increased in the brains of G2019S mice. Likewise, we saw a similar increase in the basal level of the related protein, PGC-1β. Additionally, in both WT and G2019S brains, LRRK2 inhibition caused a further increase in PGC-1β, suggesting a potential activation of mitochondrial biogenesis. In confirmation, we observed an increase of the master regulator of mitochondrial biogenesis, TFAm (*Ljubicic et al., 2010*), which coordinates the simultaneous expression of both mitochondrial and

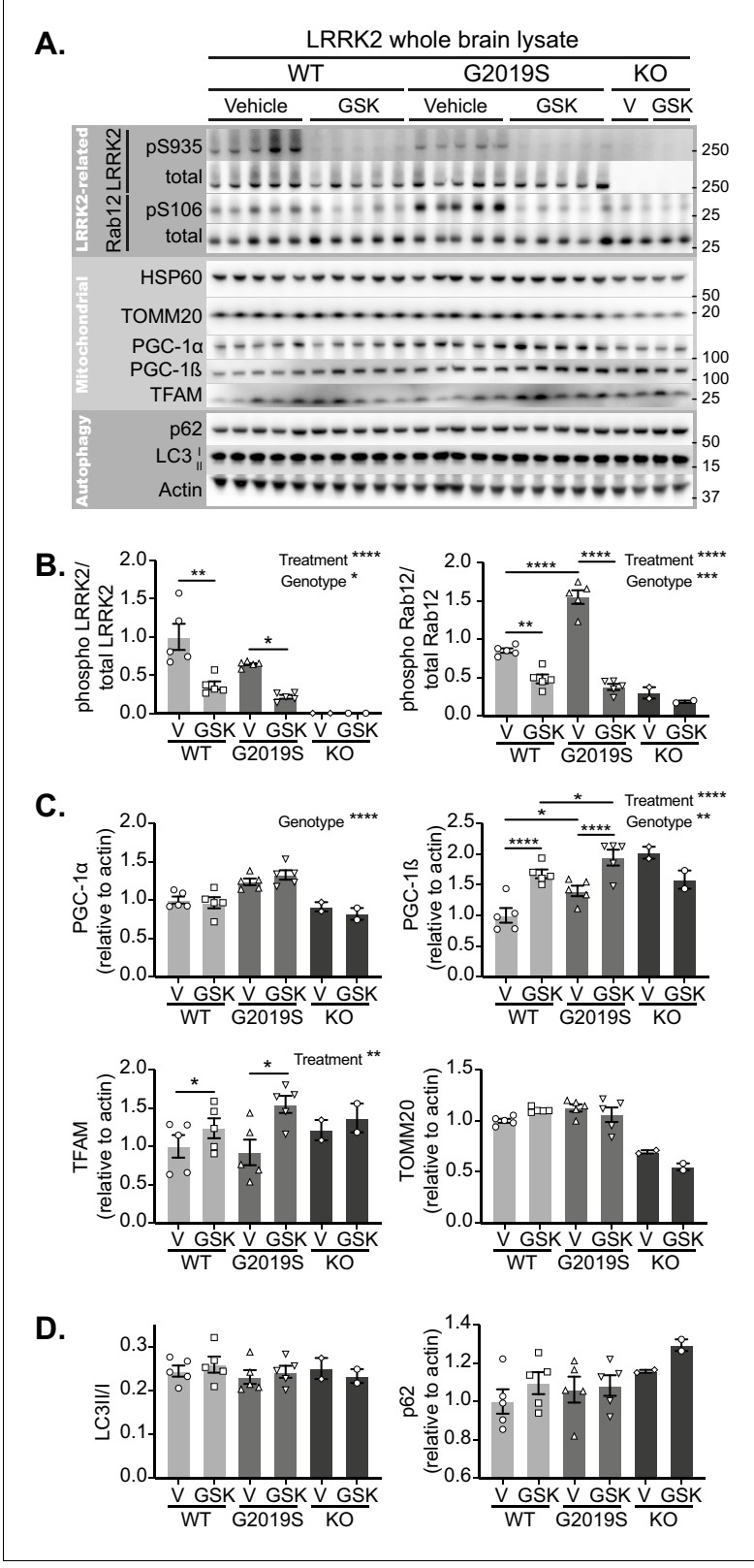

**Figure 5.** GSK3357679A treatment inhibits LRRK2 in the brain. (**A**) Immunoblots of the indicated LRRK2-related, mitochondrial and autophagy proteins from brain lysates of LRRK2 WT, LRRK2 G2019S, and LRRK2 KO *mito*-QC mice treated with vehicle or GSK3357679A. (**B**) Quantitation of phosphorylation data from the LRRK2-related proteins displayed in A. (**C**) Quantitation of mitochondria- and mitochondrial biogenesis-related proteins
*Figure 5 continued on next page*

*Figure 5 continued*

displayed in A. (D) Quantitation of autophagy-related proteins displayed in A. Overall data is represented as mean +/- SEM. Statistical significance is displayed as *p<0.05, **p<0.01, ***p<0.001, and ****p<0.0001.

The online version of this article includes the following source data and figure supplement(s) for figure 5:

**Source data 1.** Numerical data for *Figure 5*.
**Source data 2.** Western blot raw data files for *Figure 5A*.
**Figure supplement 1.** Pharmacological inhibition of LRRK2 kinase in the lung and kidney.
**Figure supplement 1—source data 1.** Numerical data for *Figure 5—figure supplement 1*.
**Figure supplement 1—source data 2.** Western blot raw data files for *Figure 5—figure supplement 1B*.

nuclear genomes, and controls the mtDNA copy number (*Figure 5A and C*). While the timecourse of GSK3357679A treatment is likely too short to see significant mitochondrial biogenesis (*Singh et al., 2015*; *Daussin et al., 2012*), it possible that this is a cellular response to changing mitophagy. While further work will be needed to validate this, a careful balance between mitochondrial turnover and biogenesis has been shown to occur previously (*Palikaras et al., 2015*). In addition to brain, we also confirmed LRRK2 kinase inhibition in the lungs and kidneys of the same animals, with GSK3357679A-treated mice showing significant loss of both LRRK2 S935 phosphorylation and Rab 10 T73 phosphorylation (*Figure 5—figure supplement 1B and C*).

As LRRK2 kinase activity was successfully inhibited in GSK3357679A-treated animals, we next analysed mitophagy. In DA neurons of the SNpc, we found that treatment with GSK3357679A increased mitophagy in both WT and G2019S KI mice. In-line with earlier results, the vehicle dosed G2019S group displayed significantly lower mitophagy compared to the vehicle- dosed WTs (*Figure 6A and B*). Importantly, treatment with GSK3357679A restored G2019S mitophagy to base-line WT levels (*Figure 6A and B*). No difference was observable in the KO groups in presence of GSK3357679A, showing mitophagy effects are through on-target LRRK2 inhibition.

As we had found differences in cortical microglial mitophagy between genotypes, we next investigated the effect of GSK3357679A treatment on this cell population. GSK3357679A increased mitophagy in the cortical microglia in both WT and G2019S groups (*Figure 4E and F*). Consistently, GSK3357679A restored G2019S mitophagy levels to a value similar to that detected in the control group (WT-V). Mitophagy was unaffected by GSK3357679A in the KO groups, confirming GSK3357679A specificity on LRRK2 kinase activity. As seen earlier (*Figure 3G*), microglial cell numbers were increased in the cortex of vehicle-treated G2019S mice compared with vehicle-treated WT mice (*Figure 6—figure supplement 1A*). Intriguingly, GSK3357679A treatment recovered the increase in number of cortical microglia observed in LRRK2 G2019S KI mice (*Figure 4—figure supplement 1B*), suggesting that LRRK2 kinase activity is a key contributor to regulation of microglial numbers in this mouse line.

In the lungs, we found that GSK3357679A increased mitophagy levels in both WT and G2019S KI animals (*Figure 6E and F*). GSK3357679A had no effect on mitophagy levels in the lungs of LRRK2 KO mice (*Figure 6F*). Importantly, in the G2019S group, GSK3357679A elevated mitophagy levels to a value similar to the WT Vehicle group, suggesting LRRK2-inhibition can also rescue the G2019S-mediated defect in mitophagy in lung. As observed for other LRRK2 inhibitors we observed enlarged lamellar bodies in Type-II pneumocytes in the lungs of mice treated with GSK3357679A, similar to that observed in LRRK2 KO mice (*Figure 4A*) and to what has been previously reported in the presence of other LRRK2 kinase inhibitors (*Fell et al., 2015*; *Baptista et al., 2018*; *Fuji et al., 2015*). Consistent with KO mice, in the kidney we found that GSK3357679A had a minimal effect on mitophagy despite this organ exhibiting robust LRRK2 inhibition (compare *Figure 4C and D* with *Figure 6—figure supplement 1B and C*). Although as previously mentioned, the very high levels of mitophagy in this tissue could be masking any subtle mitophagy increases.

In summary, these results show that a pathogenic mutation of LRRK2 impairs basal mitophagy not just in isolated cells, but also in distinct cell types within tissues. Importantly, this phenotype can be rescued by the use of LRRK2 kinase inhibitors.

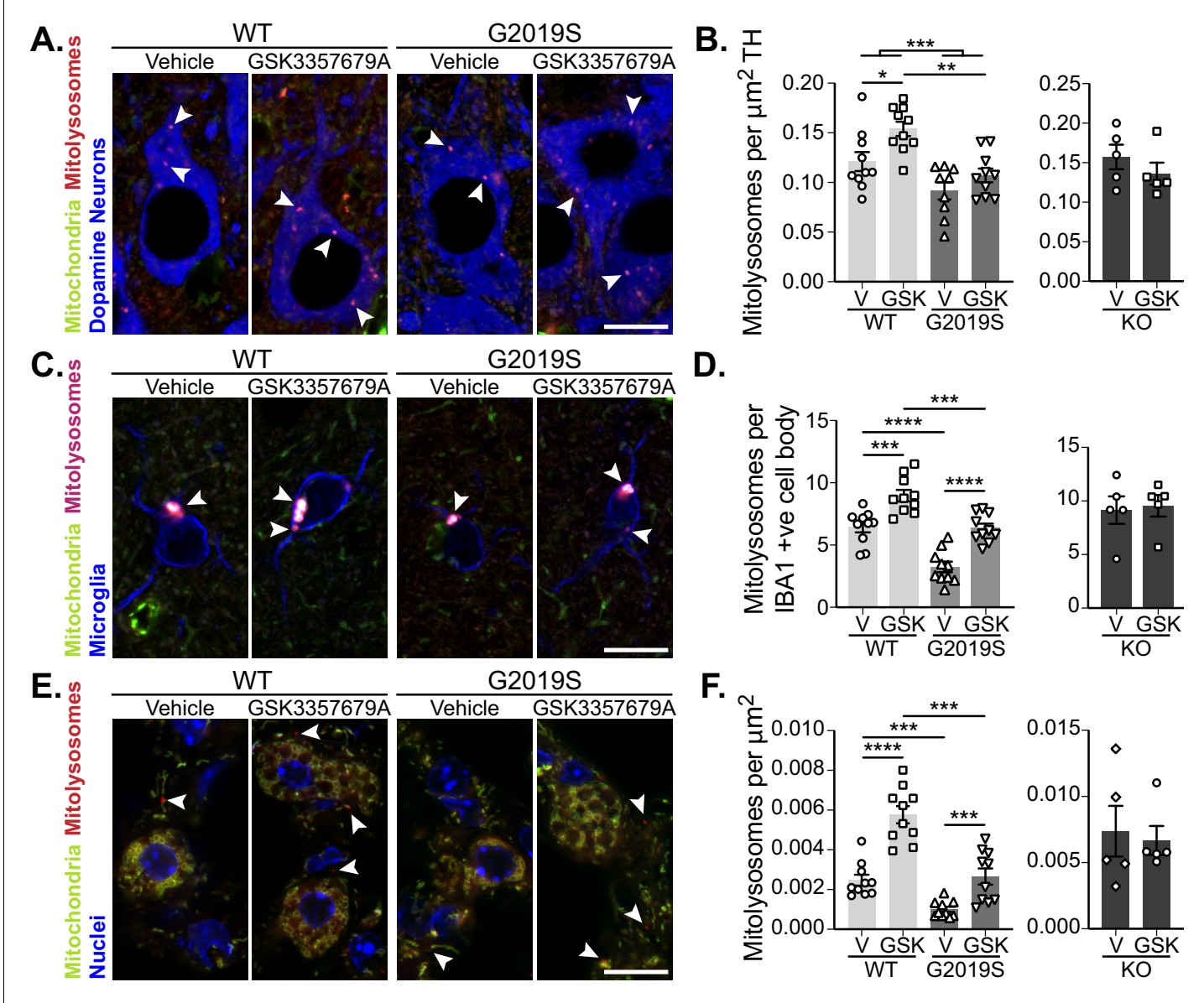

**Figure 6.** Pharmacological rescue of LRRK2-mediated mitophagy defects in vivo. (**A**) Representative image of tyrosine hydroxylase (TH) immunolabelled dopaminergic neurons of the substantia nigra pars compacta in LRRK2 WT, LRRK2 G2019S, and LRRK2 KO *mito*-QC mice treated or not (vehicle) with GSK3357679A. Arrowheads indicate mitolysosome examples. (**B**) Quantitation of mitophagy from data shown in A, with the addition of LRRK2 KO. Each data point represents mean value from an individual mouse. (**C**) Representative images of Iba1 positive cortical microglia from LRRK2 WT, and LRRK2 G2019S mice treated or not (vehicle) with GSK3357679A. Arrowheads indicate mitolysosomes. (**D**) Quantitation of mitophagy from data shown in E, with the addition of LRRK2 KO. (**E**) Representative images of *mito*-QC lungs from LRRK2 WT, and LRRK2 G2019S mice treated or not (vehicle) with GSK3357679A. Arrowheads highlight mitolysosomes. (**F**) Quantitation of mitophagy from data shown in E, with the addition of LRRK2 KO. For (**B**), (**D**), and (**F**), V = vehicle dosed animals, GSK = GSK3357679A dosed animals. Each data point represents mean value from an individual mouse. Scale bars, 10 μm. Overall data is represented as mean +/- SEM. Statistical significance is displayed as *p<0.05, **p<0.01, ***p<0.001, and ****p<0.0001. The online version of this article includes the following source data and figure supplement(s) for figure 6:

**Source data 1.** Numerical data for *Figure 6*.
**Figure supplement 1.** LRRK2 inhibition rescues microglia number in G2019S cortex but does not impact mitophagy in the kidney.
**Figure supplement 1—source data 1.** Numerical data for *Figure 6—figure supplement 1*.

## Discussion

Our work reveals that LRRK2 kinase activity inversely correlates with basal mitophagy levels, both in vitro and in vivo in specific cells and tissues. Strikingly, the mitophagy defects seemed to be specific to certain cell types. Indeed, we observed different effects in two different neuronal subpopulations and in two microglial subpopulations. More work is needed to understand why mitophagy is more sensitive to LRRK2 kinase activity in these cells, but it may help explain why DA neurons degenerate in PD. Interestingly, we found a higher mitochondrial content in the soma of Purkinje neurons compared to DA neurons, indicative of a higher oxidative metabolism in the former. In addition, mitophagy levels were much lower in the Purkinje cells, implying an inverse correlation between oxidative metabolism and mitophagy, similar to previous observations in different muscle subtypes (*Montava-Garriga et al., 2020*). Speculatively, the higher basal level of mitophagy in these DA neurons could be required to maintain oxidative metabolism in light of their lower mitochondrial numbers, thus rendering them susceptible to defects affecting mitophagy. Also of potential relevance, we observed a higher difference in mitophagy levels between genotypes in the cortical microglia than in the DA neurons. Brain resident microglia have been shown to have much higher LRRK2 levels and activity than neuronal populations, which could have implications for disease aetiology (*Schapansky et al., 2015*).

The work presented here shows for the first time that pathogenic LRRK2 mutations can alter basal mitophagy in clinically relevant cell populations in vivo. However, the extent to which impaired mitophagy drives an individual's Parkinson's disease remains to be determined. PINK1 and Parkin-mediated mitophagy is a significant pathway with respect to PD and so it was important to confirm if this was also occurring here. However, we found that loss of PINK1 (and hence a block in Parkin activation) did not impact mitophagy induced upon LRRK2 inhibition and nor did it alter the magnitude of Parkin-dependent mitophagy that occurs following mitochondrial depolarisation. This fits well with previous data generated by us and others, which demonstrates that loss of PINK1 or Parkin activity does not significantly alter basal mitophagy rates in vivo (*McWilliams et al., 2018a*; *McWilliams et al., 2018b*; *Lee et al., 2018*; *Wrighton et al., 2021*). We take this to mean that the PINK1/Parkin pathway drives mitophagy under distinct types of extreme mitochondrial stress, such as that caused during repeated exhaustive exercise (*Sliter et al., 2018*), in contrast to the basally regulated LRRK2 pathway described here. Regardless, if loss of stress-induced PINK1/Parkin-dependent mitophagy can lead to PD, then it is reasonable to assume that loss of LRRK2-regulated basal mitophagy could also contribute. These data now imply that impaired mitophagy may be a common theme in PD pathology.

Recent reports in other models support our conclusion of a role for LRRK2 in regulating mitophagy. In vitro assays in patient derived fibroblasts bearing G2019S or R1441C LRRK2 variants are consistent with our in vitro cell assays and observations in vivo (*Wauters et al., 2020*; *Bonello et al., 2019*; *Korecka et al., 2019*; *Hsieh et al., 2016*).

The mechanism by which LRRK2 kinase activity regulates basal mitophagy is currently unclear and various potential pathways exists, as discussed in our recent review (*Singh and Ganley, 2021*). LRRK2 has been shown to phosphorylate a subset of Rab GTPases (*Steger et al., 2016*) and, given the roles of Rabs in membrane trafficking, it is tempting to suggest that they may be key in regulating this mitophagy pathway (*Pfeffer, 2018*; *Pfeffer, 2017*). It has been recently shown that lysosomal overload stress induces translocation of Rab7L1 and LRRK2 to lysosomes (*Eguchi et al., 2018*). This leads to the activation of LRRK2 and the stabilisation of Rab8 and Rab10 through phosphorylation. Another recent study showed that LRRK2 mutations inhibit the mitochondrial localisation of Rab10 (*Wauters et al., 2020*). LRRK2 also plays a role in trafficking and may influence mitophagosome travel to lysosomes. Previous work has shown that LRRK2 activity can influence phagosome trafficking to lysosomes (*Härtlova et al., 2018*) and recently it was shown that LRRK2 can influence axonal autophagosome trafficking via the motor adaptor protein JIP4 (*Boecker et al., 2021*). Relatedly and also of relevance, LRRK2 regulation of JIP4 has been implicated in lysosome tubulation (*Bonet-Ponce et al., 2020*).

It is also possible that LRRK2 kinase activity alters mitochondrial function and indirectly affects mitophagy. Although we failed to detect an effect of LRRK2 inhibition of global cellular mitochondrial respiration, it is possible a small pool of mitochondria has altered oxygen consumption that impacts mitophagy. Indeed, deleterious mitochondrial phenotypes have been observed in LRRK2

G2019S patient-derived cells (*Delcambre et al., 2020*; *Mortiboys et al., 2010*; *Sanders et al., 2014*), although further work is needed to clarify if this is a cause or consequence of impaired mitophagy. It is important to note that GSK3357679A led to activation of mitochondrial biogenesis via PGC-1β and not PGC-1α. Although PGC-1α has been widely studied and described in the literature, the specific role of PGC-1β remains debated (*Gali Ramamoorthy et al., 2015*; *Singh et al., 2019*). Our current findings suggest that in the brain, PGC-1β-dependent mitochondrial biogenesis, triggered by LRRK2-dependent mitophagy, could act to prevent excessive loss of mitochondrial content and maintain mitochondrial homeostasis.

We used two highly similar reporter models in primary MEFs and in mice to study general autophagy and mitophagy. The use of both the *mito*-QC and the *auto*-QC reporters, in combination with selective LRRK2 kinase inhibitors, provided evidence that LRRK2 kinase activity affects mitophagy, rather than autophagy in general. The role of LRRK2 kinase activity on autophagy has been previously investigated in several studies with inconclusive or contradictory effects (*Plowey et al., 2008*; *Härtlova et al., 2018*; *Schapansky et al., 2014*; *Gómez-Suaga et al., 2012*; *Bravo-San Pedro et al., 2013*; *Manzoni et al., 2013*; *Orenstein et al., 2013*). However, with our reporter systems, we cannot entirely exclude that other selective autophagy pathways are affected. Additionally, total flux through the autophagy pathway is likely to be much higher than the relative flux attributable to mitophagy, and combined with the observed higher inter-individual variability with our autophagy reporter, this makes it potentially more difficult to pick up small changes. For these reasons, it would be unreasonable to entirely exclude the involvement of LRRK2 in general autophagy, although our results lack support for this.

Our results show that three structurally distinct selective LRRK2 kinase inhibitors are active on mitophagy in MEF cells, and that in vivo the tool compound GSK3357679A demonstrates similar cell-specific effects in DA neurons and microglia within the brain. Importantly, use of this inhibitor in vitro and in vivo supported our genetic data in suggesting that LRRK2 kinase activity inversely correlates with the level of basal mitophagy. The fact that we could rescue G2019S-impared mitophagy in PD-relevant cell types, within the brain, provides an exciting prospect that LRRK2 inhibitor-mediated correction of mitophagic defects in Parkinson's patients could have therapeutic utility in the clinic. In addition, LRRK2 kinase activity inhibitors could also provide a way to increase mitophagy in general, which could be beneficial in idiopathic PD, or indeed, in other non-related conditions where increased clearance of mitochondria could be beneficial, such as mitochondrial diseases. As with other LRRK2 inhibitors, GSK3357679A also induced enlarged lamellar bodies in the lung and it is possible that this effect could limit the therapeutic approach of targeting LRRK2. However, it is important to note that extensive preclinical studies in vivo failed to detect any effects of LRRK2 inhibition on lung function (*Baptista et al., 2020*) and likewise, no deleterious effects have been reported for LRRK2 kinase inhibitors in clinical studies for Parkinson's disease thus far (*Biogen Denali Therapeutics Inc, 2021a*; *Biogen Denali Therapeutics Inc, 2021b*).

Here we demonstrate, through both genetic manipulation and pharmacology, that the most common mutation in PD impairs basal mitophagy in tissues and cells of clinical relevance. The fact that we can rescue this genetic defect in mitophagy using LRRK2 inhibitors, holds promise for future PD therapeutics.

## Materials and methods

### Animals

Experiments were performed on mice genetically altered for Leucine-rich repeat kinase 2 (LRRK2), using either wild-type, LRRK2 G2019S (*Steger et al., 2016*) mutation knock-in mice, or mice in which LRRK2 has been ablated (*Parisiadou et al., 2009*; *Lin et al., 2009*) (KO).The mitophagy (*mito*-QC) and the autophagy (*auto*-QC) reporter mouse models used in this study were generated as previously described (*McWilliams et al., 2018a*; *McWilliams et al., 2016*).

### Primary mouse embryonic fibroblasts culture

Cells used in this study were Mouse Embryonic Fibroblasts (MEFs) derived in-house and were tested mycoplasma negative (MycoAlert, Lonza, LT07-318). Primary mouse embryonic fibroblasts were derived, from time-mated pregnant females at E12.5 for the LRRK2-related lines and E17.5 for

the PINK1 KO and Parkin overexpressing MEFs. Primary MEFs were maintained in DMEM (Gibco, 11960–044) supplemented with 10% FBS, 2 mM L-Glutamine (Gibco, 2503–081), 1% Na-Pyruvate (Gibco, 11360–070), 1% Non-essential amino acids (Gibco, 11140–035), 1% Antibiotics (Penicillin/Streptomycin 100 U/ml penicillin and 100 µg/ml streptomycin; Gibco), and 150 µM β-Mercaptoethanol (Gibco, 21985–023) at 37°C under a humidified 5% $CO_2$ atmosphere.

### Immortalised mouse embryonic fibroblasts

Immortalised MEFs (*McWilliams et al., 2018a*) were maintained in DMEM (Gibco, 11960–044) supplemented with 10% FBS (20% for Parkin over-expressing MEFs), 2 mM L-Glutamine (Gibco, 2503–081), 1% Na-Pyruvate (Gibco, 11360–070), 1% Non-essential amino acids (Gibco, 11140–035), 1% Antibiotics (Penicillin/Streptomycin 100 U/ml penicillin and 100 µg/ml streptomycin; Gibco), at 37°C under a humidified 5% $CO_2$ atmosphere. ATG5 knock-out immortalised MEFs and their corresponding wild-type were a kind gift from the Mizushima lab (*Kuma et al., 2004*). Cells were transduced to express the *mito*-QC reporter, and selected as previously described (*Allen et al., 2013*).

### Primary and immortalised mouse embryonic fibroblasts treatments

To assess mitophagy and autophagy upon stimulation, cells were treated for 24 hr with either 1 mM 3-Hydroxy-1,2-dimethyl-4(1H)-pyridone (Deferiprone/DFP, Sigma-Aldrich, 379409), or incubated in Earl's balanced salt solution (EBSS, Gibco, 24010–043). *mito*-QC MEFs were also treated for 24 hr with LRRK2 kinase activity inhibitors GSK2578215A (*Reith et al., 2012*) (250, 500, 1000 nM), MLi-2 (*Fell et al., 2015*) (5, 10, and 20 nM), or GSK3357679A (compound 39 (*Tasegian et al., 2021*; Ding, 2021, in preparation), 0.1, 1, 10, 100, 1000 nM). All treatments (apart from EBSS) were in DMEM (Gibco, 11960–044) supplemented with 10% FBS (20% for Parkin over-expressing MEFs), 2 mM L-Glutamine (Gibco, 2503–081), 1% Non-essential amino acids (Gibco, 11140–035), 1% Antibiotics (Penicillin/Streptomycin 100 U/ml penicillin and 100 µg/ml streptomycin; Gibco), and 150 µM β-Mercaptoethanol (Gibco, 21985–023) at 37°C under a humidified 5% $CO_2$ atmosphere. MLi-2 and GSK2578215A were synthesised by Natalia Shpiro (University of Dundee) as described previously (*Reith et al., 2012*; *Fell et al., 2015*).

### Light microscopy

MEFs were plated on glass coverslips and treated as described in the previous paragraph. At the end of the treatment, cells were washed twice in DPBS (Gibco, 14190–094), and fixed in 3.7% Paraformaldehyde (Sigma, P6148), 200 mM HEPES, pH=7.00 for 20 min. Cells were washed twice with, and then incubated for 10 min with DMEM, 10 mM HEPES. After a wash with DPBS, nuclei were stained or not with Hoechst 33342 (1 µg/mL, Thermo Scientific, 62249) for 5 min. Cells were washed in DPBS and mounted on a slide (VWR, Superfrost, 631–0909) with Prolong Diamond (Thermo Fisher Scientific, P36961). Images were acquired using a Nikon Eclipse Ti-S fluorescence microscope with a 63x objective.

### Quantitation of mitophagy and autophagy in vitro

Quantification of red-only dots was semi-automatised using the *mito*-QC counter plugin on FIJI as previously described (*Montava-Garriga et al., 2020*; *Schindelin et al., 2012*). Note, in primary MEFs we observed heterogeneity between cells and observed changes in mitophagy over increasing passages, which were uniform across *Lrrk2* genotypes. To keep quantitation consistent, we ran our experiments in sets of the same passage (simultaneously with the different genotypes) and set a threshold above which the cells were considered as mitophagic. We analysed the distribution of mitolysosomes per cell of untreated WT cells and set the threshold based on a truncated mean to remove outliers (note, outliers were only removed to set the threshold and all data were included in the quantitation). Therefore, due to the heterogeneity of the primary MEFs and passage, representing the data in percentage of mitophagic cells allowed us to better compare data sets.

Autophagosomes and autolysosomes were quantified using the Autophagy counter plugin on FIJI developed in house, following the same principle as the mito-QC counter (*Singh et al., 2020*). The macro 'auto-QC_counter.ijm' ('version 1.0 release', DOI: 10.5281/zenodo.4158361) is available from the following github repository: https://github.com/graemeball/autoQC_counter copy archived at swh:1:rev:789f01e896eb25607809239a894010e5e25c25c6 *Ball, 2020*.

## ATPB immunostaining in auto-QC primary MEFs and quantitation

Primary auto-QC MEFs were fixed as described in the previous paragraph. Cells were then permeabilised with 0.3% Triton X-100 in PBS for 5 min and washed twice in PBS/BSA 1%, followed by a 15-min incubation in PBS/BSA 1% on a shaker. Cells were incubated with the primary antibody directed against ATPB (1/200, Abcam, ab14730) prepared in PBS/BSA 1% for 1 hr at 37°C. Coverslips were washed three times in PBS/BSA 1% on shaker for 10 min, and cells were incubated with a Goat anti-Mouse IgG (H+L) Cross-Adsorbed Secondary Antibody, Pacific Blue (1/500, ThermoFisher Scientific, P31582) prepared in PBS/BSA 1% for 30 min at room temperature. Coverslips were then washed three times in PBS on shaker for 10 min and mounted as described above. Images were acquired using a Zeiss LSM880 with Airyscan laser scanning confocal microscope (Plan-Apochromat 63x/1.4 Oil DIC M27) using the optimal parameters for acquisition (Nyquist). 10–15 images were acquired per sample. Images were analysed with the Volocity Software (version 6.3, Perkin-Elmer). Images were first filtered using a fine filter to suppress noise. Subcellular structures of interest were detected by thresholding each channel. Colocalisation of autophagosomes with mitochondria was evaluated by intersecting the thresholded yellow (both red and green) signal with the thresholded signal from the blue channel. Colocalisation of autolysosomes with mitochondria was evaluated by intersecting thresholded red-only signal (identified by identified by detecting the high-intensity objects in the red channel and applying a red/green ratio threshold) with thresholded signal from the blue channel. Experiments were realized in triplicate using two passages of two MEF lines derived from independent matings. The percentage of ATPB positive autophagosomes or autolysosomes relative to their corresponding total number of autophagosomes/autolysosomes was determined by dividing the number of ATPB-colocalising autophagosomes/autolysosomes detected per field by the number of total autophagosomes/autolysosomes detected per field, multiplied by 100.

## Transmission electron microscopy (TEM) sample preparation

Cells were fixed on the dish in 4% paraformaldehyde and 2.5% glutaraldehyde in 0.1 M sodium cacodylate buffer (pH 7.2) for 30 min then scraped and transferred to a tube and fixed for a further 30 min prior to pelleting. The pellets were given three washes in cacodylate buffer, cut into small pieces and they were then post-fixed in 1% OsO4 with 1.5% Na ferricyanide in cacodylate buffer for 60 min. After another three washes they were contrasted with 1% tannic acid and 1% uranyl acetate. The cell pellets were then dehydrated through alcohol series into 100% ethanol, changed to propylene oxide left overnight in 50% propylene oxide 50% resin and finally embedded in 100% Durcupan resin (Sigma). The resin was polymerised at 60°C for 48 hr and sectioned on a Leica UCT ultramicrotome. Sections were contrasted with 3% aqueous uranyl acetate and Reynolds lead citrate before imaging on a JEOL 1200EX TEM using an SIS III camera.

## TEM quantitation

Cells for quantitative imaging were selected by uniform random sampling, ie. a random start point on the section was selected and cells chosen at uniform distances across the section. Cells were viewed initially at low magnification (×2500) and regions with electron-lucid cytoplasm were chosen and imaged at high magnification (×40,000). The images were then anonymised and scored blinded for autolysosomes with or without mitochondria. Three independent experiments were quantified with a total of 181 and 256 images per Control and GSK3357679A treated group, respectively. For each image, up to 10 autolysosomes were scored (with an average of 1.93 autolysosomes per picture). For each image, the percentage of autolysosomes containing mitochondria was calculated by dividing the number of autolysosomes containing mitochondria by the total number of autolysosome and multiplying this value by 100. The relative increase in mitolysosomes was calculated by setting the control value of each independent experiment at one to highlight the consistent increase with GSK3357679A.

## High-resolution respirometry in immortalised MEFs

Mitochondrial respiration was studied in digitonin-permeabilised cells (10 μg / $10^6$ cells) to keep mitochondria in their architectural environment. The analysis was performed in a thermostated oxygraphic chamber at 37°C with continuous stirring (Oxygraph-2 k, Oroboros instruments, Innsbruck, Austria). Cells were collected with trypsin and placed in MiR05 respiration medium (110 mM sucrose,

60 mM lactobionic acid, 0.5 mM EGTA, 3 mM MgCl2, 20 mM taurine, 10 mM $KH_2PO_4$, 20 mM HEPES adjusted to pH 7.1 with KOH at 30°C, and 1 g/l BSA essentially fatty acid free). Acute effects of GSK3357679A on the mitochondrial respiratory chain (n=4) were determined by successively injecting incremental doses of the compound (0.1 nM, 1 nM, 10 nM, 100 nM, and 100 nM) into the oxygraph chamber after activating the NS pathways (Glutamate (G) 10 mM, Malate (M) 2 mM, ADP (P) 2.5 mM, and Succinate (S) 10 mM). Chronic effects of GSK3357679A on mitochondrial respiration (n=4) were determined after 24 hr incubation with 10 nM GSK3357679A using the Substrate-Uncoupler-Inhibitor titration protocol number 2 (SUIT-002) (*Doerrier et al., 2018*). Briefly, after residual oxygen consumption in absence of endogenous fuel substrates (ROX, in presence of 2.5 mM ADP) was measured, fatty acid oxidation pathway state (F) was evaluated by adding malate (0.1 mM) and octanoyl carnitine (0.2 mM) (OctM$_P$). Membrane integrity was tested by adding cytochrome c (10 µM) (OctMc$_P$). Subsequently, the NADH electron transfer-pathway state (FN) was studied by adding a high concentration of malate (2 mM, OctM$_P$), pyruvate (5 mM, OctPM$_P$), and glutamate (10 mM, OctPGM$_P$). Then succinate (10 mM, OctPGMS$_P$) was added to stimulate the S pathway (FNS), followed by glycerophosphate (10 mM, OctPGMSGp$_P$) to reach convergent electron flow in the FNSGp-pathway to the Q-junction. Uncoupled respiration was next measured by performing a titration with CCCP (OctPGMSGp$_E$), followed by inhibition of complex I (SGp) with rotenone (0.5 µM, SGp$_E$). Finally, residual oxygen consumption (ROX) was measured by adding Antimycin A (2.5 µM). ROX was then subtracted from all respiratory states, to obtain mitochondrial respiration. Results are expressed in $pmol \cdot s^{-1} \cdot 10^6$ cells.

## Animal studies

Initial power calculations were undertaken using a two-sample, two-sided equality calculation with power set at 0.8 and type I error at 5%. Based on pilot data in the heart (mean of 60.6 mitolysosomes per section with a standard deviation of 19.8), to be able to detect a 40% change (i.e. a mean of ~24 mitolysosomes) we would require a sample size ~10. Experiments were performed on 81 adult mice (9–23 weeks old) of both genders (n=8–12 per group for the mito-QC reporter, and n=9–15 per group for the auto-QC reporter), all homozygous for the corresponding reporter (mitophagy or autophagy).

The effect of the CNS penetrant LRRK2 kinase inhibitor GSK3357679A in vivo was assessed using 50 adult mice (9–17 weeks old at the end of the study), all homozygous for the mito-QC reporter. Mice of both genders were randomly assigned to the vehicle or to the GSK3357679A-treated group (WT-Vehicle: n=10, WT-GSK3357679A: n=10, G2019S-Vehicle: n=10, G2019S-GSK3357679A: n=10, KO-Vehicle: n=5, and KO-GSK3357679A: n=5). Vehicle-treated animals were dosed (10 mL/kg) with aqueous methylcellulose (1% w/v, Sigma, M0512) prepared in sterile water (Baxter, UKF7114), or with GSK3357679A (15 mg/kg/dose) prepared in aqueous methylcellulose. Treatment was administered by oral gavage every 12 hr for a total of four times per mouse. Mice were culled 2 hr (+/- 9 min) after the last dosing.

Animals were housed in sex-matched littermate groups of between two and five animals per cage in neutral temperature environment (21° ± 1°C), with a relative humidity of 55–65%, on a 12:12 hr photoperiod, and were provided food and water ad libitum. All animal studies were ethically reviewed and carried out in accordance with the Animals (Scientific Procedures) Act 1986 as well as the GSK Policy on the Care, Welfare and Treatment of Animals, and were performed in agreement with the guidelines from Directive 2010/63/EU of the European Parliament on the protection of animals used for scientific purposes. All animal studies and breeding were approved by the University of Dundee ethical review committee, and further subjected to approved study plans by the Named Veterinary Surgeon and Compliance Officer and performed under a UK Home Office project license in agreement with the Animal Scientific Procedures Act (ASPA, 1986).

## Sample collection

Mice were terminally anesthetised with an intraperitoneal injection of pentobarbital sodium (Euthatal, Merial) then trans-cardially perfused with DPBS (Gibco, 14190–094) to remove blood. Tissues were collected and either snap frozen in liquid nitrogen and stored at −80°C for later biochemical analyses or processed by overnight immersion in freshly prepared fixative: 3.7% Paraformaldehyde (Sigma, P6148), 200 mM HEPES, pH=7.00. The next day, fixed tissues were washed three times in

DPBS, and immersed in a sucrose 30% (w/v) solution containing 0.04% sodium azide until they sank at the bottom of the tube. Samples were stored at 4°C in that sucrose solution until further processing.

## Immunolabeling of brain free-floating sections

The brain was frozen-sectioned axially using a sledge microtome (Leica, SM2010R), and 50-μm-thick sections were stored in PBS at 4°C until further treatment. Free-floating sections were permeabilised using DPBS (Gibco, 14190–094) containing 0.3% Triton X-100 (Sigma Aldrich, T8787) three times for 5 min. Sections were then blocked for 1 hr in blocking solution (DPBS containing 10% goat serum (Sigma Aldrich, G9023), and 0.3% Triton X-100). Primary antibody incubation was performed overnight in blocking solution containing one of the following antibodies: Anti-Tyrosine Hydroxylase (1/1000, Millipore, AB152), Anti-Iba-1 (1/1000, Wako, 019–19741), Anti Calbindin-D28k (1/1000, Swant, CB38), Anti-Glial Fibrillary Acidic Protein (1/1000, Millipore, MAB360). The next day, sections were washed two times for 8 min in DPBS containing 0.3% Triton X100 and then incubated for 1 hr in blocking solution containing the secondary antibody (1/200, Invitrogen P10994 Goat anti-Rabbit IgG (H+L) Cross-Adsorbed Secondary Antibody, Pacific Blue, or Invitrogen P31582 Goat anti-Mouse IgG (H+L) Cross-Adsorbed Secondary Antibody, Pacific Blue). Sections were then washed two times for 8 min in DPBS containing 0.3% Triton X100 and mounted on slides (Leica Surgipath X-tra Adhesive, 3800202) using Vectashield Antifade Mounting Medium (Vector Laboratories, H-1000) and sealed with nail polish.

## Tissue section and immunostaining

Tissues were embedded in an O.C.T. compound matrix (Scigen, 4586), frozen and sectioned with a cryostat (Leica CM1860UV). Twelve microns sections were placed on slides (Leica Surgipath X-tra Adhesive, 3800202), and then air dried and kept at −80°C until further processing. Sections were thawed at room temperature and washed three times for 5 min in DPBS (Gibco, 14190–094). Sections were then counterstained for 5 min with Hoechst 33342 (1μg/mL, Thermo Scientific, 62249). Slides were mounted using Vectashield Antifade Mounting Medium (Vector Laboratories, H-1000) and high-precision cover glasses (No. 1.5H, Marienfeld, 0107222) and sealed with transparent nail polish.

## Tissue confocal microscopy

Confocal micrographs were obtained by uniform random sampling using either a Zeiss LSM880 with Airyscan, or a Zeiss 710 laser scanning confocal microscope (Plan-Apochromat 63x/1.4 Oil DIC M27) using the optimal parameters for acquisition (Nyquist). 10–15 images were acquired per sample, depending on the tissue, by an experimenter blind to all conditions. High-resolution, representative images were obtained using the Super Resolution mode of the Zeiss LSM880 with Airyscan.

## Quantitation of mitophagy and autophagy in vivo

Quantification of mitophagy and autophagy was carried out on at least 10 pictures per sample. Images were processed with Volocity Software (version 6.3, Perkin-Elmer). Images were first filtered using a fine filter to suppress noise. Tissue was detected by thresholding the Green channel. For the immunolabelings in the brain (TH, Iba1, Calbindin D-28k, and GFAP), each cell population of interest was detected by thresholding the Pacific Blue labelled channel. A ratio image of the Red/Green channels was then created for each image.

For the mito-QC reporter, mitolysosomes were then detected by thresholding the ratio channel as objects with a high Red/Green ratio value within the tissue/cell population of interest. The same ratio channel threshold was used per organ/set of experiments. To avoid the detection of unspecific high ratio pixels in the areas of low reporter expression, a second red threshold was applied to these high ratio pixels. This double thresholding method provides a reliable detection of mitolysosomes as structures with a high Red/Green ratio value and a high Red intensity value.

For the general autophagy reporter, high-intensity red pixels were detected by thresholding the red channel within the tissue/cell population of interest. The same red channel threshold was used per organ/set of experiments. Autophagosomes and autolysosomes were then differentiated by thresholding the high-intensity red pixels depending on their Red/Green ratio channel value. Pixels

with a low Red/Green ratio were considered as autophagosomes, whereas pixels with a high Red/Green ratio were considered as autolysosomes. The same ratio channel threshold was used per organ/set of experiments. Representative 3D Isosurface Rendering were generated using the Imaris software (Bitplane, version 8.1.2).

## Filipin staining

Frozen fixed tissue sections were washed in PBS to remove any excess O.C.T compound (Scigen, 4586) excess. Sections were then incubated for 2 hr at room temperature with filipin (200 µg/mL; Sigma-Aldrich, F9765) and then washed twice in PBS. Tissue sections were mounted using Vectashield Antifade Mounting Medium (Vector Laboratories, H-1000) and sealed with nail polish. High-resolution, representative images were obtained using the Super Resolution mode of the Zeiss LSM880 with Airyscan (Plan-Apochromat 63x/1.4 Oil DIC M27).

## Western blotting

Frozen tissue was homogenised with a Cellcrusher (Cellcrusher, Cork, Ireland) tissue pulveriser. Approximately 20–30 mg of pulverised tissue were then lysed on ice for 30 min with (10 µL/mg tissue) of RIPA buffer [50 mM Tris–HCl pH 8, 150 mM NaCl, 1 mM EDTA, 1% NP-40, 1% Na-deoxycholate, 0.1% SDS, and cOmplete protease inhibitor cocktail (Roche, Basel, Switzerland)], phosphatase inhibitor cocktail (1.15 mM sodium molybdate, 4 mM sodium tartrate dihydrate, 10 mM β-glycerophosphoric acid disodium salt pentahydrate, 1 mM sodium fluoride, and 1 mM activated sodium orthovanadate), and 10 mM DTT.

For culture cells, cells were washed twice with DPBS before being lysed on ice for 10 min (60 uL per 2 ml of cell culture) with either RIPA or IP lysis buffer (50 mM HEPES pH 7.4, 150 mM NaCl, 1 mM EDTA, 10% glycerol, 0.5% NP40), cOmplete protease inhibitor cocktail (Roche, Basel, Switzerland), phosphatase inhibitor cocktail (1.15 mM sodium molybdate, 4 mM sodium tartrate dihydrate, 10 mM β-glycerophosphoric acid disodium salt pentahydrate, 1 mM sodium fluoride, and 1 mM activated sodium orthovanadate), and 10 mM DTT.

After lysis, the mixture was vortexed and centrifuged for 10 min at 4°C at 20,817xg. The supernatant was collected, and the protein concentration determined using the Pierce BCA protein assay kit (ThermoFisher Scientific, Waltham, MA, USA). For each sample, 20–25 µg of protein was separated on a NuPAGE 4–12% Bis-Tris gel (Life technologies, Carlsbad, CA, USA). Proteins were electroblotted to 0.45 µm PVDF membranes (Imobilon-P, Merck Millipore, IPVH00010; or Amersham Hybond, GE Healthcare Life Science, 10600023), and immunodetected using primary antibodies directed against phospho-Ser935 LRRK2 rabbit monoclonal (1/1000, MRC PPU Reagents and Services, UDD2), LRRK2 rabbit monoclonal (1/1000, MRC PPU Reagents and Services, UDD3), phospho-Rab10 (Thr73) rabbit monoclonal (1/1000, Abcam, ab230261), Rab10 mouse monoclonal (1/1000, nanoTools 0680–100/Rab10-605B11), recombinant Anti-Rab12 (phospho S106) rabbit antibody [MJF-R25-9] (1/1000, Abcam ab256487), Total Rab12 sheep antibody (1 µg/mL, MRC PPU Reagents and Services, SA227), PGC-1α mouse monoclonal (4C1.3) (1/1000, Millipore, ST1202), PGC-1β mouse monoclonal (E-9) (1/500, Santa Cruz Biotechnology, sc-373771), TFAm mouse monoclonal [18G102B2E11] (1/1000, abcam, ab119684), HSP60 rabbit polyclonal (D307) (1/1000, Cell Signalling Technology, #4870S), TOMM20 rabbit polyclonal (FL-145) (1/1000, Santa Cruz Biotechnology, sc-11415), p62/SQSTM1 mouse monoclonal (2C11) (1/1000, Abnova, H00008878-M01), LC3A/B rabbit polyclonal (1/1000, Cell Signalling Technology, #4108S), phospho-ubiquitin (Ser65) rabbit monoclonal (E2J6T) (1/1000, Cell Signalling Technology, #62802S), Ubiquitin (P4D1) mouse monoclonal (1/1000, BioLegend, 646302), Parkin mouse monoclonal (1/2000, Santa-Cruz Biotechnology, sc-32282), α-Tubulin (11H10) Rabbit monoclonal antibody (1/10000, CST, 2125S), and β-Actin mouse monoclonal antibody (1/1000, Proteintech, 60008–1-Ig). All antibodies to LRRK2 were generated by MRC PPU Reagents and Services, University of Dundee (http://mrcppureagents.dundee.ac.uk).

## Statistics

Data are represented as means ± SEM. Number of subjects are indicated in the respective figure legends. Statistical analyses were performed using a one-way analysis of variance (ANOVA) or two-way ANOVA followed by a Tukey HSD using RStudio version 1.1.1335 (*R Studio Team, 2015*). Statistical significance is displayed as * p< 0.05: ** p < 0.01, *** p<0.001, and **** p<0.0001.

## Acknowledgements

We acknowledge Paul Appleton at the Dundee Imaging Facility, Dundee. The Zeiss LSM880 with Airyscan was supported by the 'Wellcome Trust Multi-User Equipment Grant' [208401/Z/17/Z]. We would also like to acknowledge Dr Jin-Feng Zhao and Dr Thomas McWilliams for their expert technical assistance. This work was funded by a grant from the Medical Research Council, UK (IGG; MC_UU_00018/2) and GlaxoSmithKline. Requests for provision of GSK3357679A should be directed to Alastair Reith (alastair.d.reith@gsk.com).

## Additional information

### Funding

| Funder | Grant reference number | Author |
| --- | --- | --- |
| Medical Research Council | MC_UU_00018/2 | Ian G Ganley |

The funders had no role in study design, data collection and interpretation, or the decision to submit the work for publication.

### Author contributions

Francois Singh, Conceptualization, Formal analysis, Investigation, Visualization, Methodology, Writing - original draft, Writing - review and editing; Alan R Prescott, Investigation, Writing - review and editing; Philippa Rosewell, Formal analysis, Investigation, Writing - review and editing; Graeme Ball, Software, Methodology, Writing - review and editing; Alastair D Reith, Conceptualization, Formal analysis, Funding acquisition, Writing - review and editing; Ian G Ganley, Conceptualization, Resources, Formal analysis, Supervision, Funding acquisition, Investigation, Methodology, Writing - original draft, Project administration, Writing - review and editing

### Author ORCIDs

Francois Singh https://orcid.org/0000-0002-1696-9815
Alan R Prescott https://orcid.org/0000-0002-0747-7317
Philippa Rosewell https://orcid.org/0000-0002-1399-8602
Graeme Ball http://orcid.org/0000-0002-6526-2306
Ian G Ganley https://orcid.org/0000-0003-1481-9407

### Ethics

Animal experimentation: All animal studies were ethically reviewed and carried out in accordance with Animals (Scientific Procedures) Act 1986 as well as the GSK Policy on the Care, Welfare and Treatment of Animals, and were performed in agreement with the guidelines from Directive 2010/63/EU of the European Parliament on the protection of animals used for scientific purposes. All animal studies and breeding were approved by the University of Dundee ethical review committee, and further subjected to approved study plans by the Named Veterinary Surgeon and Compliance Officer and performed under a UK Home Office project license in agreement with the Animal Scientific Procedures Act (ASPA, 1986).

### Decision letter and Author response

Decision letter https://doi.org/10.7554/eLife.67604.sa1
Author response https://doi.org/10.7554/eLife.67604.sa2

## Additional files

### Supplementary files

- Transparent reporting form

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
