## [Decision Letter]

[Editors' note: this paper was reviewed by Review Commons.]

**Acceptance summary:**

The autophagic turnover of mitochondria is termed mitophagy. This study provides the first in vivo evidence that pathogenic, Parkinson's disease-associated LRRK2 kinase directly impairs basal mitophagy, and demonstrates that pharmacological inhibition of LRRK2 is a rational mitophagy-rescue approach and potential Parkinson's disease therapy.

---

## [Author Response]

Reviewer #1 (Evidence, reproducibility and clarity (Required)):In this manuscript, Singh et al., have directed their scientific inquiry towards understanding the role of LRRK2 (Leucine Rich Repeat Kinase 2) in mediating mitophagy, a process of clearing damaged mitochondria by autophagosomes. The discovery of PINK1 and PARK2 as Parkinson's Disease (PD) – relevant genes, has led to a significant amount of interest in delineating the role of PINK1/Parkin in mitophagy, especially induced by toxins including CCCP. However, the role of PINK1/Parkin in basal mitophagy, not induced by toxins, has been unclear. Recently, LRRK2 has emerged as a potential modulator of mitophagy. In the manuscript, the authors study the impact of LRRK2 on basal mitophagy using three mouse genotypes: WT, KO, and the pathogenic G2019S LRRK2 variant with increased kinase activity. While similar studies have been conducted on fibroblasts, PD patient-derived tissues, etc, this is the first study to recapitulate the role of G2019S LRRK2 in suppressing mitophagy in vivo. Each genotype is also engineered with a mito-QC or an auto-QC reporter at a specific nuclear locus in order to differentiate between mitochondria or autophagosomes trafficked to lysosomes for degradation, vs cytoplasmic organelles, respectively. These mouse models have been already well established by this group. Using these models, the authors have shown that the number of mitolysosomes is lower in the overactive G2019S variant and higher in the LRRK2 KO, suggesting that mitophagy is inversely correlated with LRRK2 kinase activity. After establishing these results in MEFs, the authors proceed to confirm this phenotype in different mice tissues, including four brain regions (only dopaminergic neurons and microglia show significant changes in mitophagy based on genotype), lungs, and kidneys. They proceed to test a LRRK2-specific kinase inhibitor that is CNS penetrant, GSK3357679A (GSK), and report the rescue of mitolysosomal number in brain and lung tissue, but not kidney cells of the G2019S variant. They also employ immunoblotting for the detection of LRRK2 self-phosphorylation, and Rab10 substrate phosphorylation as means of indirectly confirming the phenotypes of the mouse models and the inhibition of LRRK2 kinase activity by GSK.While the results presented are interesting, the study is largely descriptive and does not provide mechanistic insight into mitophagic vs autophagic phenotypes, nor tissue selective data. This is a real limitation to enthusiasm. What is provided is generally rigorously performed. In addition, an examination of LRRK2 expression across tissue types and cell types, as well as markers other than Rab10 would have been informative. It is promising to note that LRRK2 kinase activity inhibition could be a potential therapeutic target. However, further analysis would be required to confirm the lack of off-target or long-term effects of GSK administration – which compound has off-target effects MLi2 or the GSK, for example. How does one explain the effects of MLi2 on mitophagy in LRRK2 KO cells? There are additional concerns that need to be addressed. The various suggestions for improving the manuscript are listed below.

We thank the reviewer for their careful and considered review. They have raised important points and the suggestions below, which have tried to address, give an important mechanistic insight that was lacking from the original version of the manuscript.

Major comments:• In Figures 1C, 1H, and S1, why has the percentage of mitophagic cells been calculated as opposed to the number of mitolysosomes per cell, while in Figure 1F the number of autolosysomes per cell is presented? It is also unclear how mitophagic cells are selected – what is the specific cutoff for the number of mitolysosomes in each cell to qualify as "mitophagic cell"? Please report the mitolysosomes per cell, as that provides crucial information regarding increased mitophagy within each cell and not just within a population of cells. It will also maintain consistency of representation in other figures (number of mitolysosomes quantified by area and number of mitolysosomes per cell body). If it is not possible to report it, please include an explanation.

We apologise for the lack of consistency here between the reporters in terms of quantitation and have now displayed data for both as a percentage of mitophagic and autophagic cells. The reason for this is that in the primary *mito*-QC MEFs (but not *auto*-QC) we saw small increases in mitophagy after each passage, which were unform across all genotypes/treatments. However, as the rates of basal mitophagy are relatively low, when means were compared between individual experiments this reduced the significance. We have added the following to the description in the Methods section:

Line 488: “Note, in primary MEFs we observed heterogeneity between cells and observed changes in mitophagy over increasing passages, which were uniform across *Lrrk2* genotypes. […] Therefore, due to the heterogeneity of the primary MEFs and passage, representing the data in percentage of mitophagic cells allowed us to better compare data sets.”

• Line 430: "GSK had no effect on mitophagy levels in the lungs of LRRK2 KO mice (Figure 4G)". The statement is not consistent with the reported data as Figure 4G only shows WT and G2019S LRRK2 lungs, and not LRRK2 KO. Data not shown?

The quantitation for KO mitophagy and GSK is shown in the new Figure 6F and we will ensure this is clearly stated in the text to avoid confusion:

Line 355: “GSK3357679A had no effect on mitophagy levels in the lungs of LRRK2 KO mice (Figure 6F).”

• The authors report that MLi-2, a more potent LRRK2 inhibitor than GSK, results in decreased mitophagy at higher concentrations and note that the LRRK2-independent effects of MLi-2 on mitophagy. It raises the question whether GSK has similar side-effects as well? Are there much higher concentrations of GSK which are detrimental to the effect of mitophagy due to over-inhibition of LRRK2 kinase activity? Also, are there adverse effects of GSK administration beyond 36 hours? How will this translate to the potential use of GSK as a therapeutic tool? More experimentation comparing the two is warranted. Also, if conclusions are to be drawn/mentioned from MLi2, then the data should be present in the body of the manuscript and not relegated to supplemental data.

We would like to clarify that there are two ‘GSK’ compounds here – GS2578215 (a selective cellular tool LRRK2 benzamide compound – Reith et al., 2012) and the novel structurally distinct GSK3357679A that is more potent and selective (Ding et al., in prep). We apologise if were not clear in the text, as the potency of the novel GSK3357679A is similar to that of MLi-2, and using a 50-fold higher concentration (compared to MLi-2) no side-effects were observed in the LRRK2 KO cells (Figure 1). Some confusion may have arisen due the use of the previously published and less potent GSK2578215A in the same figure as MLi-2 (S1). We have clarified and simplified the message in the text (as shown below). Our aim in this work was not to provide a direct comparison between MLi-2 and GSK3357679A, but rather show that other LRRK2 inhibitors also cause increased mitophagy – and that the specific compound used for the in vivo studies gave results similar to other LRRK2 inhibitors in the cell-based assay. In terms of in vivo tolerability profile of GSK3357679A in rodents, we have bid orally dosed GSK3357679A for up to 14 days with no adverse effects (Ding et al., in prep). We currently have no data beyond 14 days bid oral dosing in mice, but have no reason to anticipate any adverse effects. The 36 hour dosing duration used for the mitophagy work reported here was selected solely in consideration of animal use ethics – as we observed the effect of compound on mitophagy within this relatively short dosing window.

Line 136: “To further support a role for LRRK2 kinase activity in negatively regulating basal mitophagy we utilised two additional and structurally distinct tool LRRK2 kinase inhibitors, GSK2578215A ^26^ and MLi-2 ^27^, in primary *mito*-QC MEFs. […] Thus, genetically and chemically, the data show that LRRK2 inhibition enhances basal mitophagy in cells and in these assays, GSK3357679A displayed a superior performance compared to other available LRRK2 kinase inhibitors.”

• What is the mechanism of GSK in inducing mitophagy? The authors have previously reported that basal mitophagy does not require PINK1/Parkin (McWilliams et al., 2018). Is the increased mitophagy by GSK dependent on PINK1/Parkin? Recently Bonello et al., 2019 have reported that LRRK2 impairs basal mitophagy in a Parkin-dependent manner, that can be rescued by the LRRK2 kinase inhibitor LRRK2-IN-1. The comparison of the effect of GSK on mitolysosomes between WT, Pink1 KO, Parkin KO, and PINK1 KO + LRRK2 G2019S variant would be insightful in dissecting the mode of action of GSK and contribute towards the understanding of the players involved in basal (non-toxin induced) mitophagy.

We thank the reviewer for raising this point on mechanism, which is something we are committed to working out. While it will be beyond the scope of the current manuscript to work out the complete mechanism of LRRK2-mediated mitophagy, we have addressed the important point raised by the Reviewer. Using previously generated primary PINK1 KO mito-QC MEFs (McWilliams et al., 2018) we have shown that the GSK3357679A-induced mitophagy is independent of PINK1 and we detected no changes in phosphoubiquitin levels due to LRRK2 inhibition. These data are shown in a new Figure 2G-I. Additionally, we carried out the classical Parkin-dependent assays and found that LRRK2 inhibition had no effect on this (new Figure S2D-F). The Bonello et al., paper is an interesting manuscript and we reference it in our work here – however, the mechanistic aspect of this work involves Parkin overexpression and thus we feel that they are examining a different pathway. In summary, we feel that the work here is monitoring a PINK1 (and hence Parkin)-independent pathway.

• Additionally, the fact that PINK1/Parkin was not found essential for basal mitophagy in mice (McWilliams et al., 2018) and *Drosophila* (Lee et al., 2018), but is essential in human cells (Bonello et al., 2019) and is not trivial when it comes to the clinical relevance of GSK as a pharmacological inhibitor of LRRK2 to induce mitophagy and should be addressed in the Discussion section. In addition, this section will greatly benefit from discussing the opposing data and its ramifications in the area of LRRK2-induced basal mitophagy.

We would like to point out that PINK1/Parkin mitophagy is not essential for all pathways in human cells – indeed we have previous published that mitophagy can be induced independently of Parkin in primary human PD patient fibroblasts (and other human-derived cell lines, Allen et al. 2013, PMID: 24176932). There has also been an additional manuscript published showing that basal mitophagy occurs independently of PINK1 in Zebrafish (Wrighton et al., 2021, PMID: 33536245). Also, from the work by Sliter et al. looking at mitophagy induced in vivo following exhaustive exercise, though the mitophagy that was induced was dependent on PINK1, the basal level of mitophagy was not – hence similar to our reported results. We do not feel our results conflict with the vast majority of work, nor do we feel it controversial to show that there are mitophagy pathways that are independent of PINK1/Parkin. Regardless of any PINK1/Parkin involvement, we do not think this takes away from the potential that LRRK2-mediated modulation of mitophagy may contribute to disease risk and potential treatment. We have added to the Discussion section, as noted here.

Line 369: “PINK1 and Parkin-mediated mitophagy is a significant pathway with respect to PD and so it was important to confirm if this was also occurring here. […] Regardless, if loss of stress-induced PINK1/Parkin-dependent mitophagy can lead to PD, then it is reasonable to assume that loss of LRRK2-regulated basal mitophagy could also contribute”.

• Brain Rab 10?

We and other independent groups at our institution have been unable to detect high levels of phosphorylated Rab10 in brain extracts. We cannot fully explain this, but it may be possible that a Rab10 phosphatase is highly active in this region. However, we have now found that another LRRK2 substrate, Rab12, is readily phosphorylated in brain and provides an additional marker for LRRK2 activity. These new data is included in a new Figure 5.

• While this manuscript focuses on mitophagy, it fails to address any potential effects of LRRK2 on mitochondrial biogenesis. How is the total mitochondrial number and mitochondrial biogenesis affected by the LRRK2 genotypes and GSK application? For example, is an increase or decrease in mitophagy compensated by a concomitant decrease or increase in mitochondrial biogenesis, or does it remain the same? Tools such as MitoTimer used in conjunction with mitoQC and autoQC would yield significant information regarding the biological mechanisms underlying LRRK2's effect on mitochondrial homeostasis. Not required, just a mechanistic suggestion.

The Reviewer raises a very important point here and we thank them for this suggestion. We believe the overall level of mitophagy under basal conditions is relatively low compared to the total mitochondrial pool, however over time this could lead to significant differences in mitochondrial number if biogenesis is not compensated for. To look at mitochondrial content and biogenesis markers, we probed brain tissue lysates in the different *Lrrk2* genotypes treated with/without GSK3357679A. Surprisingly, we found that LRRK2 inhibition did indeed trigger upregulation of mitochondrial biogenesis markers (PGC1B and TFAM). The new data is shown in Figure 5.

The text has been changed to discuss this.

Line 301: “Of potential relevance, we found that the mitochondrial biogenesis marker PGC-1α was increased in the brains of G2019S mice. […] While further work will be needed to validate this, a careful balance between mitochondrial turnover and biogenesis has been shown to occur previously ^43^.”

To further look at mitochondrial function, we examined their ultrastructure by TEM and also performed high-resolution respirometry in isolated MEFs, via the Oroboros Oxygraph-2k. LRRK2 inhibition did not globally alter mitochondrial function. This new data is shown in Figure S2I and H and discussed in the text.

Line 190: “…we noticed no obvious changes to mitochondrial morphology and ultrastructure, as observed using TEM (Figure S2G). Secondly, using high resolution respirometry, we measured mitochondrial oxygen consumption. […] However, given that the fraction of mitochondria targeted for mitophagy is likely small compared to the total pool, we cannot rule out that this population is functionally impaired.”

Minor comments:• In Figure S1, what is the effect of DFP on autolysosomal number? Similarly, what is the effect of serum starvation on mitophagy?

We have these data and apologise for their omission – LRRK2 genotypes do not alter mitophagy/autophagy levels under these stimuli. These data are now included in the revised Figure S1.

• Line 305: please expand PK/PD.

Expanded to Pharmacokinetics/pharmacodynamics.

• Line 424: Figures 4C and 4D show dopaminergic neurons and not lungs as mentioned in the text. Figure legends mention that lungs are shown in 4G and 4H.• In-text citation of Figure 4H is missing.• Figure S1D, p-value indications are missing in the figure, although mentioned in the figure legend.

Apologies – we have corrected these mistakes/omissions in the text.

• Line 430: "Consistent with the genetics, in the kidney we found that GSK had a minimal effect on mitophagy despite this organ exhibiting robust LRRK2 inhibition". It is not clear how the lack of increase in mitolysosome number in kidneys is consistent with genetics. Please explain or rephrase.

We have rephrased the text as we agree it is confusing – we were referring to the genetics as in the LRRK2 KO animals.

Line 617: “Consistent with KO mice, in the kidney we found that GSK3357679A had a minimal effect on mitophagy despite this organ exhibiting robust LRRK2 inhibition (compare Figure 4C and D with S6B and C).”

Reviewer #2 (Evidence, reproducibility and clarity (Required)):In this paper, authors provide evidences for a potential role of LRRK2 in the regulation of basal mitophagy in different tissues. LRRK2 is a protein kinase often mutated in PD. One of the most common is the G2019S kinase- gain of function mutation. Using mitophagy (mito-QC) and autophagy (auto-QC) reporter mice, authors demonstrate that basal mitophagy is regulated by LRRK2 kinase activity in vitro and ex vivo. In particular, mitophagy is downregulated in MEFs deriving from G2019S transgenic mice, and this has been confirmed ex vivo in specific neuronal cells population, and in the lungs. They also demonstrate that genetic ablation of LRRK2 or chemical inhibition of its enzymatic activity (with a highly specific kinase inhibitor named GSK3357679A-unpublished) can rescue the mitophagy defects observed in LRRK2 G2019S mice. Autophagy levels measured in cells and tissues are unaffected between the different genotypes. This finding led the authors to conclude that LRRK2 kinase activity acts specifically on basal mitophagy, rather than autophagy in general.

We thank the Reviewer for their comprehensive analysis of this manuscript and support for our findings. As described below, we have tried to address all the comments to add important mechanistic data.

Major comments:Overall, the results of the paper are convincing and clearly demonstrate a role of LRRK2 kinase activity in the regulation of basal mitophagy. Data and methods are very well described and the number of replicates and statistical analysis is adequate.It is however unexplored how LRRK2 regulates mitophagy. As such, this is a descriptive work that shows for the first time a role for LRRK2 in mitochondrial quality control, but does not propose a molecular mechanism for this very intriguing finding.The use of the mito-QC probe is an extremely valuable tool to measure basal mitophagy in vivo and in vitro, but should be ideally paralleled by other approaches. Decreased number of red mitochondria in mito-QC expressing cells/tissue can be the result of delayed acidification, which does not necessarily correlates to altered mitochondrial quality control. For this reason, we strongly advice measuring the overall effect on mitochondrial mass/content (see suggested additional experiments) and/or direct interaction between autophagic components and mitochondria.

We have carried out additional *mito*-QC independent experiments, as described below.

The effect on basal mitophagy upon GSK3357679A treatment is very interesting, and might hold therapeutic opportunities: are these mitochondria that are targeted to mitophagy dysfunctional? In other words, is this the result of mitochondrial quality control, or does it affect overall mitochondria turnover (i.e. functional mitochondria as well)? This is very important to clarify. Enhanced mitochondrial degradation can be counterproductive, unless specifically targeted to dysfunctional mitochondria and/or paralleled by increased mitochondrial biogenesis.

We thank the reviewer for this important comment and have performed additional work to look at mitochondrial function and biogenesis. Consistent with the small levels of mitophagy occurring (relative to levels that we observe following stimulation with DFP, CCCP etc.), we were unable to detect global changes in mitochondrial morphology (EM) or respiration (Oxygraph) following LRRK2 inhibition. These data are shown in a new Figure S2G-I and described in the text.

Line 190: “…we noticed no obvious changes to mitochondrial morphology and ultrastructure, as observed using TEM (Figure S2G). Secondly, using high resolution respirometry, we measured mitochondrial oxygen consumption. […] However, given that the fraction of mitochondria targeted for mitophagy is likely small compared to the total pool, we cannot rule out that this population is functionally impaired.”

We also blotted for mitochondrial content in brain tissue and again, consistent with small changes in mitophagy, we did not detect a large change in following LRRK2 inhibition at this timepoint, though the level of mitochondrial markers was less in KO animals, which is consistent with the increased mitophagy in this tissue. These data are in a new Figure 5A and C. We also followed up on the mitochondrial biogenesis aspect, and we thank the Reviewer for suggesting these experiments as they yielded relevant results that showed upregulation of signalling following LRRK2 inhibition in the brain. The new data is shown in Figure 5A and C.

Additional text has been added to discuss this:

Line 301 “Of potential relevance, we found that the mitochondrial biogenesis marker PGC-1α was increased in the brains of G2019S mice. […] While further work will be needed to validate this, a careful balance between mitochondrial turnover and biogenesis has been shown to occur previously ^43^.”

Details on the characterization of LRRK2 kinase inhibitor GSK3357679A are not available. Data are unpublished, and the manuscript is in preparation (ref: Ding., et al.). It is therefore difficult to evaluate whether GSK3357679A is doing what the authors claim, in terms of LRRK2 kinase activity-inhibition. Is this work currently under revision? Authors are proposing a back to back? Would it be possible to see the results on the characterization of this novel inhibitor?

We also appreciate the comments regarding the Ding et al., manuscript describing the characterisations of the novel GSK LRRK2 inhibitor GSK3357679A. This is a separate GSK manuscript, outside of our lab, and we had assumed this manuscript would be available at the time of review. However, detailed characterisation of the properties of GSK3357679A are now in the public domain (https://www.biorxiv.org/content/10.1101/2021.05.21.445132v2), as referenced in the manuscript. The molecular structure of GSK3357679A will be reported in the Ding et al. medicinal chemistry paper that is in the final stages of approval at GSK. It is our hope that this data will shortly be available (in time for publication of this work). Following public disclosure of the structure of GSK3357679A it is intended to make this compound openly available to the research community via a third-party chemical supply company.

Suggested additional experiments and specific comments:• Only one approach (mito-QC) has been used to measure mitophagy. It would be valuable to integrate the obtained results with additional approaches. For example by evaluating steady state levels of mitochondrial resident proteins as read out of mitochondrial content. Interaction between autophagy marker LC3 II and mitochondria via IF is also a desirable approach to investigate autophagy of mitochondria. This is doable at least for the in vitro part of the paper using MEFs cells.

The *mito*-QC assay is the most sensitive one we have in terms of monitoring mitophagy and can easily detect low levels of mitophagy, such as basal mitophagy, that could be missed using measurements of the total mitochondrial pool. However, the Reviewer makes an important and valid point to use alternate methods. We have now measured the co-localisation of mitochondrially localised ATPB and LC3 (our tandem reporter version) and did indeed find a small degree of co-localisation between mitochondria and autophagosomes as well as autolysosomes, which is consistent with the mito-QC data. We also employed EM and found that LRRK2 inhibition resulted in a significant fold-increase in mitolysosomes. These data confirm mitophagy and help rule out alternate pathways such as MDVs. These data have been incorporated into a new Figure 2A-C and E-F.

• Autophagy levels measured in cells and tissues are unaffected between the different genotypes. This finding led the authors to conclude that LRRK2 kinase activity acts specifically on basal mitophagy, rather than autophagy in general. It is important to confirm these results by Western Blot analysis of LC3 II protein. As suggested by the authors, it is possible that a block in mitophagy will have little influence on the total autophagic levels. However, it is also possible, that LRRK2 regulates mitochondrial quality control via autophagy-independent mechanisms such as those that are directly regulated by endosome-lysosome interaction (MDV or Rab5-dependent). This possibility could explain why LRRK2 regulates mitophagy without apparently impacting general autophagy.

This is an intriguing possibility that LRRK2 maybe regulating mitochondrial turnover via an autophagy-independent mechanism. In addition to the EM data mentioned above that shows partially degraded mitochondria in lysosomes, we also measured the effects of LRRK2 inhibition on mitophagy in ATG5 KO MEFs (From the Mizushima Lab). Loss of ATG5 blocked the increase in mitophagy, implying LRRK2-dependent mitophagy requires the conventional autophagy machinery. The new data are shown in Figure 2D:

• Mitophagy does not seem to be negatively affected in TH neurons of G2019S mice (Figure 2C): although a trend toward reduced mitophagy is observed, this is not significant. We suggest to rephrase "….., mitophagy appeared reduced in DA neurons of LRRK2 G2019S KI mice compared to WT (Figure 2A and C). This was similar to our earlier observations in MEFs and showed that the presence of LRRK2 can impact mitophagy in this clinically relevant population of neurons within the midbrain." The sentence is inaccurate; TH neurons from G2019S mice do not show decreased mitophagy. It is actually worrisome that LRRK2 gain-of-function mutant impairs mitophagy in MEFs cells but does not seem to have an effect on disease-relevant neurons (i.e. TH positive cells). We suggest consolidating this trend with additional biological replicates (strongly advisable), or rephrasing the paragraph accordingly.

The Reviewer is of course correct here and we apologise for the confusion, we simply wanted to state that there is a downward trend, which is clearly shown – but this is not significant. We have not increased biological replicates, given that a similar experiment was already performed with 10 mice per condition (as opposed to 9 for this one, new Figure 3 A and C). The data in the previous Figure 4D, with 10 mice per condition does show a significant genotype effect using a 2-way ANOVA – the genotype significance was unfortunately omitted from the original Figure 4D and will be shown and discussed in the revised manuscript:

Line 213 “Basal mitophagy was significantly enhanced in the LRRK2 KO neurons compared to WT, and although not statistically significant, mitophagy appeared reduced in DA neurons of LRRK2 G2019S KI mice compared to WT (Figure 3A and C). While we cannot say that mitophagy is significantly impaired in the G2019S DA neurons from this set of experiments using nine individual mice, we do however find a significant mitophagy reduction in G2019S DA neurons in a later set of experiments, with ten mice per condition (Figure 6A and B)”.

• In the analyzed mutant cells (KO and G2019S), stress (DFP)-induced mitophagy is comparable across all genotypes, which led the authors to conclude that LRRK2 predominantly influences basal mitophagy. This sounds a bit of an overstatement considering the peculiar way by which DFP induces mitophagy. DFP does not affect ΔΨm, and it appears to trigger mitophagy via PINK1/Parkin-independent mechanism. To clarify this point it would be helpful to repeat the experiment in MEFs cells using CCCP or valinomycin.

This a very good point and we analysed the classical Parkin dependent mitophagy pathway in MEFs using CCCP. LRRK2 inhibition did not alter mitophagy under any of these conditions and additionally the LRRK2-depednent mitophagy was independent of PINK1. These new data are shown in Figure S2D-F and Figure 2G-I respectively.

• In the description of figure 4C-D the authors claim that "In DA neurons of the SNpc, we found that treatment with GSK3357679A increased mitophagy in both WT and G2019S KI mice" however the figure shows that the increase of mitophagy in G2019S mice treated with GSK335779A is not significant compared to the vehicle-treated, and that should be clarified in the text. Overall, it seems that in TH neurons of G2019S mice, mitophagy is not clearly affected.

Again, apologies for the confusion and the text will be rephrased here as mentioned above. There is indeed a significant difference, via a 2-way ANOVA, in WT vs G2019S mitophagy in this experiment (but unfortunately we only showed the GSK3357679A significance in the original figure). This is displayed in the new Figure 6A and B.

Minor comments:• Quantification of mitophagy is differently expressed throughout the paper. For example:– In MEFs, it is expressed as Mitophagic cells %. Considering that the mito-QC Counter should report an increase in the total number of mitolysosomes, what are the parameters that define a mitophagic cell? Cells with at least 1 red (or more?)-only dots? This should be clarified.– In microglia, it is expressed as Mitolysosomes per cell body– In all the remaining tissues, it is presented as Mitolysosomes per areaIt is advisable to present these data in a consistent way, if possible.

We apologise for the lack of clarity/consistency here – we have unified quantitation values in MEFs as percentage of mitophagic/autophagic cells and state in the Methods as to our reasoning behind this:

Line 487: “Note, in primary MEFs we observed heterogeneity between cells and observed changes in mitophagy over increasing passages, which were uniform across *Lrrk2* genotypes. […] Therefore, due to the heterogeneity of the primary MEFs and passage, representing the data in percentage of mitophagic cells allowed us to better compare data sets.”

For microglia, we found that their numbers were increased in G2019S cortex, hence for this case we normalised to cell number. We have better explained this in the text.

Line 234: “when mitophagy quantitation was normalised for cell number (mitolysosomes per Iba1 positive cell body per field), we found a significant decrease in basal mitophagy in G2019S microglia compared to WT, as well as an increase in mitophagy levels in KO cells (Figure 3H)”.

• In Figure 4D the description in the text states that "the vehicle dosed G2019S group displayed significantly lower mitophagy compared to the vehicle- dosed WTs" The statement seems inaccurate.

As above, we have included the omitted 2-way ANOVA significance on genotype.

• typos: lane 66 Figure 1D instead of 1C.

This has been corrected.

Reviewer #3 (Evidence, reproducibility and clarity (Required)):The authors investigated the role of LRRK2, a protein kinase frequently mutated in Parkinson's disease (PD), on mitophagy in vivo. Using mitophagy and autophagy reporter mice, bearing either knockout of LRRK2 or expressing the pathogenic kinase-activating G2019S LRRK2 mutation, the authors found that basal mitophagy was specifically altered in clinically relevant cells and tissues. Basal mitophagy inversely correlates with LRRK2 kinase activity in vivo, and use of distinct LRRK2 kinase inhibitors in cells increased basal mitophagy. Indeed, CNS penetrant LRRK2 kinase inhibitor, GSK3357679A, rescued the mitophagy defects observed in LRRK2 G2019S mice. This study provides in vivo evidence that pathogenic LRRK2 directly impairs basal mitophagy, a process with strong links to idiopathic Parkinson's disease, and demonstrates that pharmacological inhibition of LRRK2 is a rational mitophagy-rescue approach and potential PD therapy.

We thank the Reviewer for all their work in constructively assessing our manuscript. As described below, we hope to have addressed all the comments in a satisfactory manner and in doing so have added clarifying mechanistic insights.

Major comments:I suggest that the authors discuss following points more.(1) The authors surmised that the failure of higher doses of MLi-2 to stimulate mitophagy is due to an off-target effect. Indeed, at 20 nM, MLi-2 inhibited mitophagy even in the LRRK2 KO cells (Figure S1D). Is it possible that TBK1 is inhibited by higher doses of MLi-2 and consequently mitophagy is decreased?

The Reviewer raises an interesting point and we have examined a previous in-house kinase screen on the specificity of MLi-2 (found here: http://www.kinase-screen.mrc.ac.uk/screening-compounds/672622 ). However, there was no identified inhibition of TBK1, even at high doses of MLi-2. Our main intention was to show that additional structurally distinct LRRK2 inhibitors also enhance mitophagy (which they do) – it is just that we found at high concentrations of MLi-2, there appeared to be an off-target effect. We have toned down the text as follows:

Line 136 “To further support a role for LRRK2 kinase activity in negatively regulating basal mitophagy we utilised two additional and structurally distinct tool LRRK2 kinase inhibitors, GSK2578215A ^26^ and MLi-2 ^27^, in primary *mito*-QC MEFs. As with GSK3357679A, both these compounds were able to inhibit LRRK2 in cells and increase mitophagy (Figure S1D and E). We do note that at high concentrations, MLi2 failed to stimulate mitophagy and this may be due to an off-target effect, as at 20 nM it also inhibited mitophagy in the LRRK2 KO cells (Figure S1D). Thus, genetically and chemically, the data show that LRRK2 inhibition enhances basal mitophagy in cells and in these assays, GSK3357679A displayed a superior performance compared to other available LRRK2 kinase inhibitors.”

(2) To use LRRK2 kinase inhibitor as a therapeutic drug against PD, it is important to suppress the cytotoxic effect derived from LRRK2 dysfunction. For example, if LRRK2 inhibition causes an abnormal phenotype in lung, it is important to avoid such abnormalities. In this work, enlarged lamellar bodies were observed in the G2019S mice when GSK3357679A elevated mitophagy levels to a value similar to the WT control (Figure 4G and 4H). Namely, when the mitophagy value is equivalent to WT mice, the lung phenotype looks similar to LRRK2 KO mice. If the authors fine-tuned the concentration of GSK3357679A, both the basal mitophagy and the pulmonary phenotype become similar to WT? Or else, the toxic phenotype in lung does not appear in patients with lrrk2 G2019K mutation? The authors need to discuss this topic to make their statement about LRRK2 inhibitor more convincing.

This is an important point and has been much discussed in the field following previous observations of the pulmonary phenotype. We included the mouse lung histopathology data to illustrate that GSK3357679A acts in a similar manner to other reported LRRK2 kinase inhibitors, in that it induces an increase in lamellar body density in type II pneumocytes in the lung in mice. Recent work though has found that extensive in vivo and ex vivo preclinical studies, funded by Michael J Fox Foundation (Baptista et al., 2020), failed to detect any effects of LRRK2 kinase inhibitors on lung function. Additionally, no deleterious effects on lung function have been reported for LRRK2 kinase inhibitors in clinical studies for Parkinson’s disease thus far. We have added this information to the discussion as follows:

Line 431: “As with other LRRK2 inhibitors, GSK3357679A also induced enlarged lamellar bodies in the lung and it is possible that this effect could limit the therapeutic approach of targeting LRRK2. However, it is important to note that extensive preclinical studies in vivo failed to detect any effects of LRRK2 inhibition on lung function ^69^ and likewise, no deleterious effects have been reported for LRRK2 kinase inhibitors in clinical studies for Parkinson’s disease thus far ^70,71^.”.

(3) page 21, line 424, "Figure 4C and 4D" should be "Figure 4G and 4E".

We apologise for these mistakes and they have been corrected in the revision.

(4) From great body of works in vitro, I agree that the LRRK2 pathogenic mutation (especially G2019S) is a kinase-activated mutation. On the other hand, no clear increase in either LRRK2 autophosphorylation or Rab10 phosphorylation was observed in lrrk2 G2019S KI mice (Figure 4A). Only Rab10 phosphorylation in kidney seems increase in lrrk2 G2019S KI mice. Is there any legitimate reason to explain these data?

Yes, this result does not follow published data showing increased kinase activity for the G2019S mutant and we cannot fully explain it. We speculate that a Rab10 phosphatase may be more active in some tissues, hence the increase in phosphorylation is not readily observed. However, we have now analysed an additional LRRK2 substrate, Rab12, and find its phosphorylation is increased in brain G2019S tissues vs WT – It is possible that specific Rab substrates are more sensitive to the G2019S mutation. We have included the new data in a revised Figure 5.

(5) The authors states that the loss of PINK1 or Parkin activity does not significantly alter basal mitophagy rates in vivo, and three papers are cited in the manuscript [Cell Metab (2018) and Open Biol (2018) by McWilliams et al., and JCB (2018) by Lee et al]. Indeed, from their own data using mito-QU mice, the authors have legitimate concerns about a current model.However, contribution of PINK1 and Parkin to mitophagy in vivo is still controversial as rather conflicting results have been reported by other three papers [Cornelissen et al., (2018) Deficiency of parkin and PINK1 impairs age-dependent mitophagy in *Drosophila*. eLife 7; Sliter et al., (2018) Parkin and PINK1 mitigate STING-induced inflammation. Nature 561; Kim et al., (2019) Assessment of mitophagy in mt-Keima *Drosophila* revealed an essential role of the PINK1-Parkin pathway in mitophagy induction in vivo. FASEB J 33]. Even if these papers focused on aging- or mitochondrial stress-dependent mitophagy, and there were methodological differences (not mito-QC but mito-Keima) to monitor mitophagy, these papers are worth citing in the Discussion section.

This is an important point raised by the Reviewer and touched on by the other Reviewers. Firstly, we have confirmed that the LRRK2-dependent pathway observed here is independent of PINK1. Using previously generated primary PINK1 KO mito-QC MEFs (McWilliams et al., 2018) we have shown that the GSK3357679A-induced mitophagy is independent of PINK1 and we detected no changes in phosphoubiquitin levels due to LRRK2 inhibition. Additionally, we carried out the classical Parkin-dependent assays and found that LRRK2 inhibition had no effect on this. These data are shown in the new Figure 2G-I and S2D-F.

We do hope that we are not being controversial here (or previously). Our past work has shown that not all mitophagy pathways go through PINK1/Parkin – and here we propose that both dependent and independent pathways could contribute to mitochondrial dysfunction and PD. We do note that recently published work using mtKeima and a tandem mCherry-GFP reporter also found that basal mitophagy is independent of PINK1 in Zebrafish (Wrighton et al., 2021, PMID: 33536245). Also, the article by Lee et al., compared both *mito*-QC and mtKeima and did not find any role for PINK1 and Parkin under basal conditions. We think that the main point is that we and others have looked under basal conditions, whilst the “conflicting” papers mentioned have looked under stress conditions. We do not think we are being controversial to suggest multiple mitophagy pathways exists. Indeed, from the work by Sliter et al. looking at mitophagy following repeated exhaustive exercise, though the mitophagy that was induced was dependent on PINK1, the basal level of mitophagy was not – hence similar to our reported results! Regardless of any PINK1/Parkin involvement, we do not think this takes away from the potential that LRRK2-mediated modulation of mitophagy may contribute to disease risk and potential treatment. To note this, we have added to the Discussion section:

Line 369: “PINK1 and Parkin-mediated mitophagy is a significant pathway with respect to PD and so it was important to confirm if this was also occurring here. However, we found that loss of PINK1 (and hence a block in Parkin activation) did not impact mitophagy induced upon LRRK2 inhibition and nor did it alter the magnitude of Parkin-dependent mitophagy that occurs following mitochondrial depolarisation. This fits well with previous data generated by us and others, which demonstrates that loss of PINK1 or Parkin activity does not significantly alter basal mitophagy rates in vivo ^17,18,46,47.^ We take this to mean that the PINK1/Parkin pathway drives mitophagy under distinct types of extreme mitochondrial stress, such as that caused during repeated exhaustive exercise ^48^, in contrast to the basally regulated LRRK2 pathway described here. Regardless, if loss of stress-induced PINK1/Parkin-dependent mitophagy can lead to PD, then it is reasonable to assume that loss of LRRK2-regulated basal mitophagy could also contribute.”

(6) Paper by Lee et al. (basal mitophagy is widespread in *Drosophila* but minimally affected by loss of pink1 or parkin. J. Cell Biol 2018) is duplicated as citation # 19 and 39.

Apologies, the duplication has been removed.